# A film-lever actuated switch technology for multifunctional, on-demand, and robust manipulation of liquids

Chao Liang [1], Zihang Yang [2] & Hanqing Jiang [1] ✉

A lab-on-a-chip system with Point-of-Care testing capability offers rapid and accurate diagnostic potential and is useful in resource-limited settings where biomedical equipment and skilled professionals are not readily available. However, a Point-of-Care testing system that simultaneously possesses all required features of multifunctional dispensing, on-demand release, robust operations, and capability for long-term reagent storage is still a major challenge. Here, we describe a film-lever actuated switch technology that can manipulate liquids in any direction, provide accurate and proportional release response to the applied pneumatic pressure, as well as sustain robustness during abrupt movements and vibrations. Based on the technology, we also describe development of a polymerase chain reaction system that integrates reagent introduction, mixing and reaction functions all in one process, which accomplishes "sample-in-answer-out" performance for all clinical nasal samples from 18 patients with Influenza and 18 individual controls, in good concordance of fluorescence intensity with standard polymerase chain reaction (Pearson coefficients > 0.9). The proposed platform promises robust automation of biomedical analysis, and thus can accelerate the commercialization of a range of Point-of-Care testing devices.

Emerging human diseases, such as the COVID-19 pandemic of 2020 that has led to the loss of millions of lives, are a major threat to global health and human civilization[1]. Early, rapid, and accurate disease detection is critical to controlling the spread of a virus and for achieving improved treatment outcomes. Mainstream diagnostic ecosystems based on centralized laboratories, where testing samples are sent to hospitals or diagnostic clinics and are operated by professional personnel, currently limit access to close to 5.8 billion people worldwide, particularly those who live in low-resource settings that lack expensive biomedical equipment and skilled clinicians[2]. The development of a low-cost and user-friendly lab-on-a-chip system with point-of-care testing (POCT) capability that provides physicians with timely diagnostic information to make informed decisions regarding diagnosis and treatment is thus highly desirable[3].

The World Health Organization (WHO) guidelines state that an ideal POCT must be affordable, user-friendly (easy to use with minimal training), accurate (avoid false-negative or false-positive results), rapid and robust (ensures good reproducibility), and deliverable (capable of long-term storage and easily obtained by end-users)[4]. To meet these requirements, POCT systems should offer the following features: multifunctional dispensing to decrease manual intervention, on-demand release to proportionally control reagent transportation for accurate testing results, and robust operations to resist vibrations from the environment. The most widely used POCT devices currently are lateral flow strips[5,6] consisting of several layers of porous nitrocellulose membranes that drive very small amounts of sample forward, while reacting to pre-immobilized reagents via a capillary force. Although they are low in cost, easy to use, and offer the advantages of

[1]School of Engineering, Westlake University, Hangzhou 310024, China. [2]Vantronics Hangzhou Intelligence Technology Ltd., Hangzhou 311100, China. ✉e-mail: hanqing.jiang@westlake.edu.cn

fast results, the flow strips-based POCT devices can only be applied to bioassays (e.g., glucose level test[7,8] and pregnancy test[9,10]) without requiring multistep reactions (e.g., multi-reagents loading, mixing, multiplexed reaction). Additionally, the driving force to control the motion of the fluid (i.e., capillary force) does not offer good coherence, especially among different batches, resulting in poor reproducibility[11] and making the lateral flow strips useful mainly for qualitative detections[12,13].

Advanced micro and nanoscale manufacturing capabilities have created opportunities for developing microfluidic-based POCT devices for quantitative measurements[14–17]. By tuning the interfacial properties[18,19] and the geometry of the channels[20–22], the capillary force and flow rate of these devices can be controlled. Their robustness especially for highly wetting liquids is still not acceptable, however, due to their fabrication inaccuracy, material imperfections, and susceptibility to environmental vibrations[23]. Moreover, since the capillary flow is generated at the liquid-air interface, additional flows cannot be introduced, particularly after the microfluidic channels are filled with liquid. As a result, several sample-introducing steps must be conducted to achieve a more complex assay[24,25].

Among microfluidic devices, the centrifugal microfluidic device currently represents one of the best POCT solutions[26,27]. Its driving mechanism is advantageous, namely, by adjusting the rotational frequency one can control the actuation forces. However, the drawback is that the centrifugal force is always directed to the outside rim of the device, creating difficulties in achieving multistep reactions that are necessary for more complex assays. Even though additional actuation forces (e.g., capillary[28,29] among many others[30–35]) are introduced in addition to the centrifugal force to achieve the multifunctional dispensing, unintentional liquid transport may still occur since these additional forces are mostly orders of magnitudes lower than the centrifugal forces, making them only effective in small operating ranges, or incapable of on-demand liquid release use. Combining pneumatic operations in centrifugal microfluidics, e.g., the centrifugo-dynamic method[36–38], the thermo-pneumatic method[39] the active pneumatic method[40] have shown to be an appealing alternative. In the contrifugo-dynamic method, an additional cavity and connecting microchannels are integrated in the device to enable both outward and inward manipulations, though its pumping efficiency (ranging from 75 to 90%) depends highly on number of pumping cycles as well as the viscosity of the fluids. In the thermo-pneumatic method, a latex membrane and a liquid transition chamber were specially designed to seal or reopen an inlet when a trapped air volume is heated or cooled. However, the heating/cooling setup brings about a slow actuation issue and restrict its usage in thermal-sensitive assay (e.g., polymerase chain reaction (PCR) amplification). In the active pneumatic method, on-demand fluid releasing and inward manipulation are achieved by simultaneously applying positive pressure and precisely matched rotational speed, enabled by a high-speed motor. There have been other successful methods that employ only pneumatic driving mechanisms (positive[41,42] or negative pressure[43]) and a normally closed valve structure. By applying pressure sequentially in the pneumatic chamber, the liquid is pumped forward in a peristaltic way by which the normally closed valve avoids the liquid from flowing back, thereby enabling complex fluid manipulations. However, currently there are only limited microfluidic technologies that can perform complex fluid handling in a single POCT device, which includes multifunctional dispensing, on-demand release, robust operation, long-term storage, high-viscosity liquid handling, and cost-effective fabrication—all at the same time. A lack of multi-step functionality operating also could be one of the reasons why only a few commercial POCT products (e.g., Cepheid, Binx, Visby, Cobas Liat, and Rhonda) to date have been successfully adopted in the open market.

In this paper, we presented a type of pneumatic microfluidic driving mechanism based on a film-lever actuated switch technology (FAST). FAST simultaneously incorporates all the necessary characteristics and is capable to handle a range of reagents, from microliter to several milliliters. FAST is comprised of an elastic film, a lever, and a block. When pneumatic pressure is not being applied, the film, lever, and block can be sealed tightly, and the liquid inside can be stored for long periods of time. When appropriate pressure that is tunable based on the length of the lever is applied, the film expands and pushes the lever open to let the liquid flow through. This allows for multifunctional dispensing of liquid in either a cascaded, simultaneous, sequential, or selective manner.

We have developed a PCR system using FAST to realize "sample-in-answer-out" results for influenza A and B virus (IAV and IBV) detection. We achieved a lower limit of detection (LOD) of $10^2$ copies/ ml and our multiplexed tests demonstrated specificity for IAV and IBV and provided pathotyping capability for the influenza virus. The clinical testing results using the nasal swab sample from 18 patients and 18 healthy individuals show good concordance in fluorescence intensity with standard RT-PCR (Pearson coefficients > 0.9). The estimated material cost of the FAST-POCT device is about $1 (Supplementary Table 1), which can be further reduced when using mass-manufacturing method (e.g., mold injection). In practical terms, the FAST-based POCT device has all the required characteristics as envisioned by the WHO, is compatible with emerging biochemical testing methods such as plasmonic thermocycle testing[44], amplification-free immunoassay[45] and nanobody-functionalized testing[46], which suggests opportunities for POCT systems.

## Results

### Design, mechanism, and theoretical analysis of FAST-POCT platform

Figure 1a illustrates the structure of a FAST-POCT platform consisting of four fluidic chambers: the pre-storage chamber, mixing chamber, reaction chamber, and waste chamber. A key enabler to control the flow of fluids is the FAST structure (consisting of an elastic film, a lever, and a block) located at the pre-storage and mixing chambers. As a pneumatic-driven method, the FAST structure allows for accurate control of fluid flow, including switching between the sealed and opened states, multifunctional dispensing, on-demand fluid release, robust operations (e.g., insensitive to environmental vibrations) and long-term storage. The FAST-POCT platform consists of four layers—substrate layer, elastic film layer, plastic film layer, and cover layer—as seen in an expanded view in Fig. 1b (also detailed in the Supplementary Figs. S1 and S2). All liquid transport channels and chambers (e.g., pre-storage and reacting chambers) are integrated on the substrate made from PLA (polylactic acid), with a thickness from 0.2 mm (thinnest part) to 5 mm. The elastic film material is PDMS, with a thickness of 300 μm and is easily expandable when air pressure is applied due to its "thin thickness" and small elastic modulus (about 2.25 MPa[47]). The plastic film layer is made from polyethylene terephthalate (PET) with a thickness of 100 μm and is used to protect the elastic film from becoming overly deformed due to pneumatic pressure. Corresponding to the chambers on the substrate layer, there are levers that connect with the cover layer (made of PLA) through hinges to control the liquid flow. The elastic film is adhered to the substrate layer by a double-sided adhesive tape (ARseal 90880) and covered with the plastic film. T-shaped clip structures are used in the cover layer to assemble the three layers on the substrate. The T-shape clip has a clearance between two legs. When the clip is pushed into the groove, the two legs bend slightly and then recover to its original state as it comes through the groove and tightly bind the cover and the substrate (Supplementary Fig. S1). A connector is then used to assemble the four layers.

The working mechanism of the FAST-POCT platform is shown in Fig. 2. The key components are a block on the substrate layer and a hinge on the cover layer, which leads to an interference fit when the

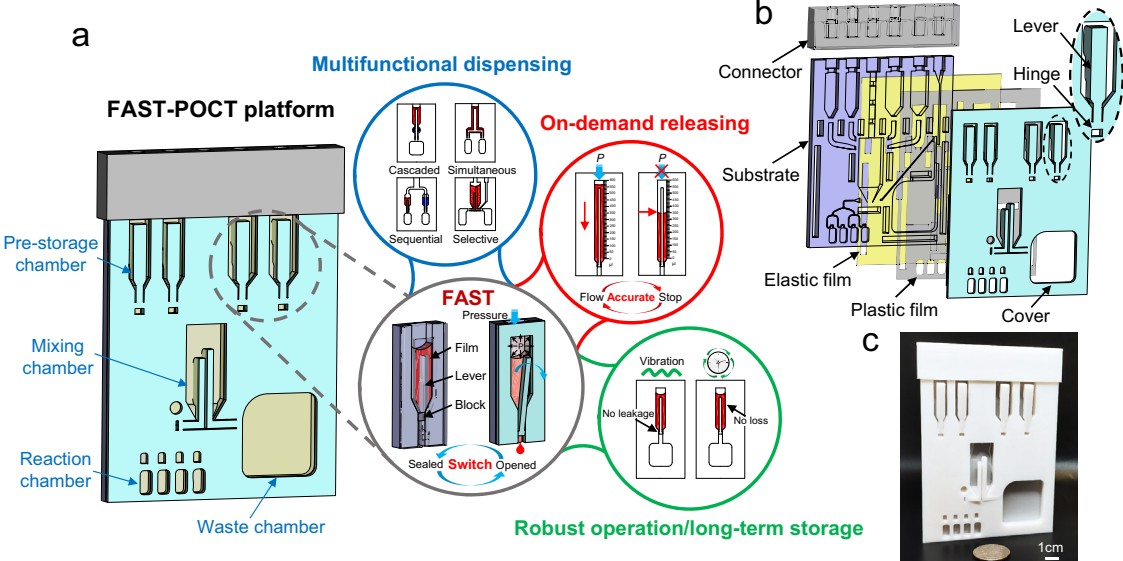

**Fig. 1 | Schematic illustration of FAST-POCT platform. a** Schematic diagram of the platform with illustration of different functional chambers and the features of FAST. **b** Expanded view schematic diagram of the FAST-POCT platform. **c** Photo of the platform next to a US quarter coin.

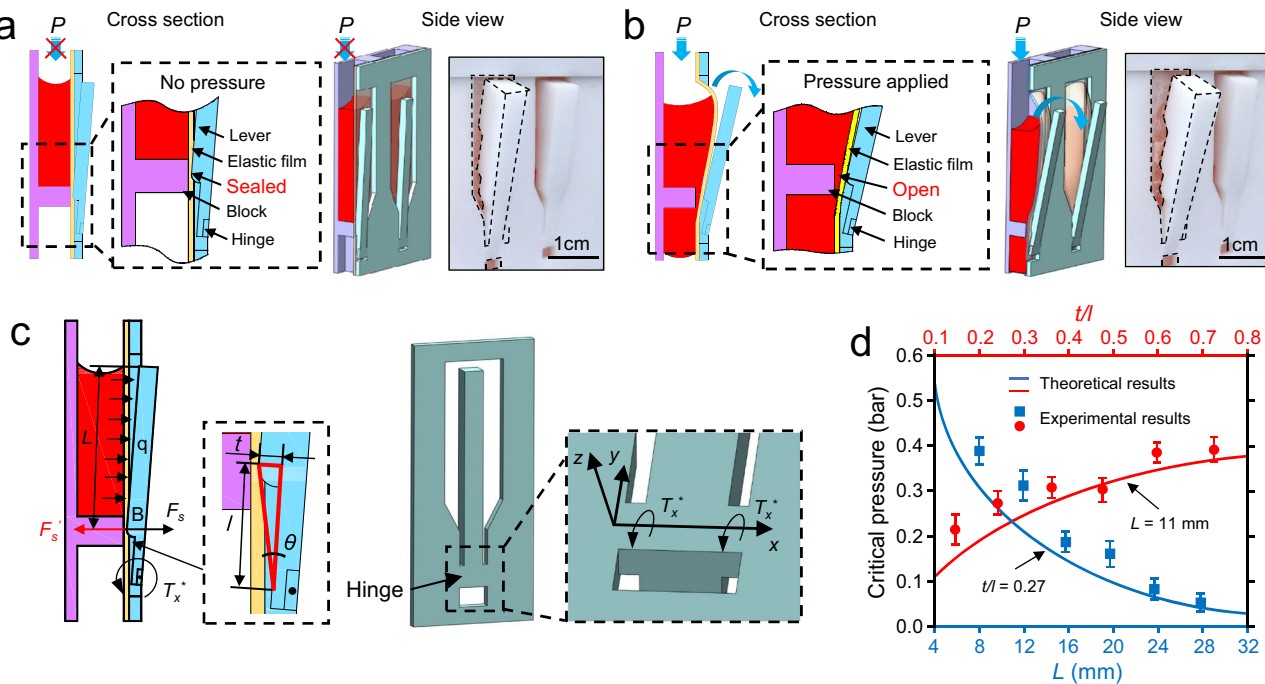

**Fig. 2 | Mechanism and theoretical analysis of FAST device.** Schematic diagrams and pictures for **a**, Sealed state. When no pressure is applied, the levers press the film to the blocks and the liquid is sealed. **b** Opened state. When pressure is applied, the film is expanded and pushes the lever up, thus, the channel opens, and the liquid can flow. **c** Characteristic sizes that determine the critical pressure. Characteristic sizes involve the length of the lever ($L$), the distance between the block and the hinge ($l$) and the protrusion thickness of the lever ($t$). $F_s$ is the sealing force at the block point B. $q$ is a uniformly distributed load on the lever. $T_x^*$ denotes the torque at the hinge generated by lever. Critical pressure means the pressure needed to push the lever up and make the liquid flow. **d** The theoretical and experimental results for the relationship between the critical pressure and the characteristic sizes. $n = 6$ independent experiments were conducted with the data shown as ± s.d. Source data are provided as a Source Data file.

four layers are assembled through the T-shaped clip structures. When the pneumatic pressure is not applied (Fig. 2a), the interference fit causes the bending deformation of the hinge, exerting the sealing forces through the lever to press the elastic film against the block and to seal the liquid in the chamber, which is defined as a sealed state. It should be noted that in this state, the lever bends outwards, as seen from the side view in Fig. 2a. When the pneumatic pressure is applied (Fig. 2b), the elastic film expands outwards to the cover layer and

pushes the lever up; thus, a gap opens between the lever and the block to allow the liquid to flow to the next chamber, which is defined as an opened state. When the pneumatic pressure is removed, the lever can recover to its original position and remain in its sealed state due to the elastic property of the hinges. The video of the movement of the lever is provided in Supplementary Movie S1.

An analytical model based on beam theory is developed as follows to analyze the critical pressure $P_c$, at which a gap is opened

as a function of the geometrical parameters (e.g., $L$ is the lever length, $l$ is the distance between the block and the hinge, $S$ is the contact area of the lever with the liquid and $t$ is the protrusion thickness of the lever shown in Fig. 2c). As detailed in the Supplementary Note and Supplementary Figure S3, a gap opens when $P_c \geq \frac{2F_s l}{Sl}$, where $F_s$ is the sealing force related to the interference fit and leads the torque $T_x^*(=F_s l)$ generated by the bending deformation of the hinge. Experimental characterizations and the analytical models exhibit good agreement (Fig. 2d), showing that critical pressure $P_c$ increases with the increasing of $t/l$ and the decreasing of $L$, which is readily explained by the classical beam model, i.e., torque increases with $t/l$. Our theoretical analysis thus clearly demonstrates that critical pressure can be effectively tuned by adjusting the lever length $L$ and $t/l$ ratio, thus providing an important basis for the design of the FAST-POCT platform.

## Performance of the FAST-POCT platform

FAST-POCT platform enables multifunctional dispensing (as shown by Fig. 3a with illustrations and experiments), which is the most important feature for a successful POCT wherein the liquid can be manipulated in any direction and in any order of either cascading, simultaneous, sequential, or selective multifunctional dispensing. Figure 3a(i) presents the cascaded dispensing mode in which two or more chambers are connected in a cascaded way using the blocks to separate the different reagents and one lever to control the open and close states. When pressure is applied, the liquid flows from the top to the bottom chamber in a cascaded manner. It should be noted that the cascaded chambers can be filled with wet chemicals or dry chemicals such as lyophilized powders. In the experiment in Fig. 3a(i), the red ink from the top chamber flowed to the second chamber with blue dye powders (copper sulfate) and became dark blue as it reached the bottom chamber. Here, the control pressure to the injected liquid is also shown. Similarly, when one lever is connected to two chambers, it becomes the simultaneous injection mode as shown in Fig. 3a(ii) in which the liquid can be equally distributed to two or more chambers when a pressure is applied. Since the critical pressure depends on the length of the lever, one can adjust the lever length to achieve a sequential injection mode, as shown in Fig. 3a(iii). A long lever (with critical pressure $P_{c\_long}$) was connected to chamber B and a short lever (with critical pressure $P_{c\_short} > P_{c\_long}$) was connected to chamber A. As pressure $P_1$ ($P_{c\_long} < P_1 < P_{c\_short}$) was applied, only the liquid in red can flow to chamber B and when the pressure was increased to $P_2$ ($> P_{c\_short}$), the blue liquid can flow to chamber A. This sequential injection mode applies to different liquids transferring to their related chambers in sequence, which is critical for a successful POCT device. Figure 3a(iv) demonstrates the selective injection mode, where the main chamber had a short (with critical pressure $P_{c\_short}$) and a long lever (with critical pressure $P_{c\_long} < P_{c\_short}$) that were connected to chamber A and chamber B, respectively, in addition to another air channel connected to chamber B. To transfer the liquid to chamber A first, pressure $P_1$ ($P_{c\_long} < P_1 < P_{c\_short}$) and $P_2$ ($P_2 > P_1$) with $P_1 + P_2 > P_{c\_short}$ were applied to the device at the same time. This way the liquid was blocked from entering chamber B by $P_2$; meanwhile, the total pressure $P_1 + P_2$ exceeded the critical pressure to activate the shorter lever connected to chamber A to allow the liquid flow to chamber A. Then, when chamber B was required to be filled, we only need to apply $P_1$ ($P_{c\_long} < P_1 < P_{c\_short}$) in the main chamber to activate the long lever and allow the liquid to flow to chamber B. It can be clearly observed from time $t = 3$ s to 9 s that the liquid in chamber A remained constant while it increased in chamber B when pressure $P_1$ was applied. When chamber A needed to be filled again, we only need to apply $P_1$ in the main chamber and $P_2$ in the additional chamber. This way, the flow behavior can switch selectively between chambers A and B. The flow behavior of the four multifunctional dispensing modes can be found in Supplementary Movie S2.

Long-term reagent storge is another essential characteristic for a successful POCT device, which will allow untrained personnel to handle multiple reagents. Although many techniques show their potential for long-term storage (e.g., micro-dispenser[35], blister[48], and stick packages[49]), a special receiving chamber is required to hold the packages, thereby increasing cost and complexity; moreover, these storage mechanisms do not enable on-demand releasing and lead to loss of reagents due to residuals in the packages. The capability of long-term storage was tested by conducting accelerated life tests, using PMMA material fabricated by the CNC technique due to its small roughness and the resistance to gas permeation (Supplementary Fig. S5). The testing devices were filled with DI-water (deionized water) and 70% ethanol (to simulate the volatile reagents) at 65 °C for 9 days. Both DI water and ethanol were stored using an aluminum foil to seal their top entrance. An Arrhenius equation and the activation energy for permeation reported in the literature[50,51] were applied for calculating the equivalent real time. Figure 3b shows the results the average weight loss of five samples maintained at 9 days at 65 °C, i.e., equivalent to 0.30% for DI water and 0.72% for 70% ethanol for over 2 years at 23 °C.

Figure 3c presents the robustness testing under vibration. Since the capillary valve (CV) is the most popular liquid manipulating technique in existing POCT devices[28,29], a CV device with a width of 300 μm and depth of 200 μm is used for comparison. It is observed that when both devices were kept still, the liquid in the FAST-POCT platform was sealed and the liquid for the CV device was pinned due to the abrupt expansion of the channel, which reduced the capillary force. However, as the angular vibrational frequency of the orbital shaker increased, the liquid in the FAST-POCT platform remained sealed, but the liquid in the CV device flowed to the bottom chamber (see also in Supplementary Movie S3). This suggests that the deformed hinge in the FAST-POCT platform can provide a robust mechanical force to the block and thus tightly seal the liquid in the chamber. However, for the CV device, the liquid is pinned owing to the equilibrium among the solid, air, and liquid phases, thereby creating instability and the potential for the vibration to break the equilibrium and result in unintended flow behavior. The advantage of the FAST-POCT platform is that it provides robust functioning and avoids operational failure in the presence of vibration, which typically occurs during delivery and operation.

Another important feature of FAST-POCT platform is its on-demand release performance, which is a critical requirement in quantitative analysis. Figure 3d compares the on-demand release for both FAST-POCT platform and a CV device. From Fig. 3d(iii) we see that the FAST device has a quick response to the pressure signal. When the pressure was applied on a FAST-POCT platform, the liquid flowed; the flow stopped immediately once the pressure was removed (Fig. 3d(i)). This action can be attributed to the quick elastic recovery of the hinge, which pushes the lever back to the block and thus seals the chamber. In the CV device, however, the liquid continues to flow, leading to approximately 100 μl unintended liquid volume at the end when the pressure is removed (Fig. 3d(ii) and Supplementary Movie S4). This can be attributed to the disappearance of the capillary pinning effect upon the full wetting of the CV subsequent to the first injection.

The capability of handling liquids with different wettability and viscosity in a same device is still challenging for the POCT application. Low wettability may cause leaking or other unintentional flowing behavior in the channel and the preparation of high viscosity liquid often requires auxiliary instrument, such as vortex mixers, centrifuges, and strainers[52]. We tested the relationship between the critical pressure and the liquid properties (with a wide range of wettability and viscosity). The results are shown in Table 1 and Movie S5. It can be seen that the liquid with different wettability and viscosity can all be sealed tightly in the chamber, and when the pressure is applied, even the liquid with the viscosity as high as 5500 cP can also be transferred the next chamber, which makes the high viscosity sample testing (i.e.,

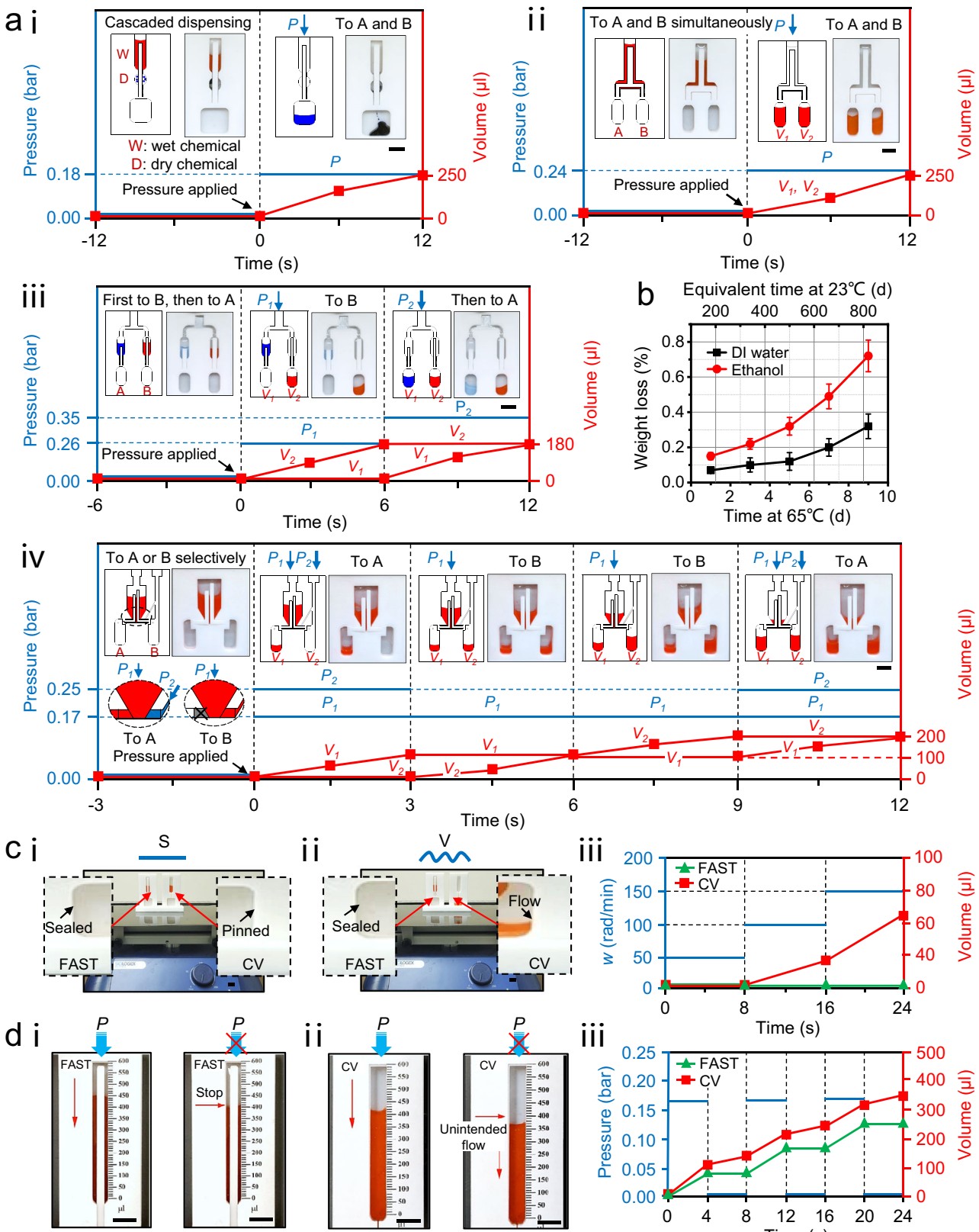

**Fig. 3 | Characterization of FAST device. a** Illustration of the multifunctional dispensing, namely (i) cascaded, (ii) simultaneous, (iii) sequential, and (iv) selective dispensing. The curves denote the working process and parameters of these four dispensing modes. **b** The results for the long-term storage tests of DI water and Ethanol. $n = 5$ independent experiments were conducted with the data shown as ± s.d. **c** Demonstration of the robustness tests when the FAST device and capillary valve (CV)-based device are in (i) static state and (ii) vibrating state. (iii) The volume verse time for FAST- and CV-devices under different angular vibrational frequency. **d** Results for on-demand releasing tests of (i) FAST-device and (ii) CV-device. (iii) The relationship between the volume and the time for FAST- and CV-devices using an intermittent pressure mode. All the scale bars, 1 cm. Source data are provided as a Source Data file.

**Table 1 | The relationship between liquid properties and critical pressure**

| Description | Acetone | Ethanol | Methanol | DI water | Glycerin | Sylgard 184 |
|---|---|---|---|---|---|---|
| Contact angle (°) | 17.2 | 22.7 | 25.3 | 87.2 | 45.6 | 62.3 |
| Viscosity (cP) | 0.3 | 1.2 | 0.8 | 1.0 | 1500.0 | 5500.0 |
| Critical pressure (bar) | 0.116 | 0.128 | 0.121 | 0.154 | 0.218 | 0.317 |

sputum, a highly viscous specimen used to diagnose respiratory disease) possible.

### Working principles of the FAST-POCT platform

By combining the above-mentioned multifunctional dispensing unit, a broad range of FAST-based POCT devices can be developed. We demonstrated one example as shown in Fig. 1. This unit contains pre-storage chambers, mixing chamber, reaction chamber, and a waste chamber. The reaction reagents can be stored long term in the pre-storage chambers, and then can be released to the mixing chamber. The mixed reagents can be selectively transferred to the waste chamber or reaction chamber when the proper pressure is applied.

Since PCR testing is the gold standard for pathogen detection (e.g., H1N1 and COVID-19) and involves multiple reaction steps, we used the FAST-POCT platform for PCR testing as an application. Figure 4 shows the PCR testing procedure using the FAST-POCT platform. The elution reagents, magnetic microbead reagents, washing solution A, and washing solution W were first pipetted into the pre-storage chambers E, M, $W_1$, and $W_2$, respectively. The RNA adsorption steps are shown in Fig. 4a and are as follows: (1) The sample was pipetted into the M chamber and released to the mixing chamber when the pressure $P_1$ (= 0.26 bar) was applied. (2) The air pressure $P_2$ (= 0.12 bar) was applied through channel A, which was connected to the bottom of the mixing chamber. Though plenty of mixing techniques show their potentials for liquid mixing in the POCT platforms (e.g., serpentine mixing[53], chaotic mixing[54] and batch-mode mixing[55]), their mixing efficiency and effectiveness are still unsatisfactory. Here, a bubble mixing method is employed, in which air is introduced to the bottom of the mixing chamber generating air bubbles in the liquid; the strong vortex then enables full mixing performance within several seconds. The bubble mixing experiments were carried out and the results are provided in Supplementary Fig. S6. It can be seen that when 0.10 bar pressure was applied, it took about 8 s to complete a full mixing. As the pressure increased to 0.20 bar, it only took about 2 s to obtain a full mixing. The method to calculate the mixing efficiency is provided in the method section. (3) A rubidium magnet was used to extract the microbeads, followed by a pressure $P_3$ (= 0.17 bar) through P channel to transfer the reagents to the waste chamber. Figure 4b, c shows the washing steps, which were performed to remove the impurities from the sample, as follows: (1) The washing A solution from $W_1$ chamber was released to the mixing chamber using pressure $P_1$. (2) The bubble mixing process was then performed. (3) The washing A solution was transferred to the waste chamber, with the microbeads extracted in the mixing chamber by the magnet. The washing W process (Fig. 4c) is similar to washing A (Fig. 4b). It should be noted that each step for washing A and W was performed two times. Figure 4d shows the elution step where the RNA is eluted from the microbeads; the elution injection and mixing steps are the same as that of the above-mentioned RNA adsorption and washing steps. Since the elution reagent was transferred to the PCR reaction chamber, pressure $P_3$ and $P_4$ (= 0.23 bar) were applied at the same time, which achieved the critical pressure of the lever for sealing the PCR reaction chamber. Similarly, the pressure $P_4$ also helps in sealing the channel that leads to the waste chamber. Thus, all the elution reagents were equally distributed to the four PCR reaction chambers for eliciting a multiplex PCR reaction. The above-mentioned process is provided in Supplementary Movie S6.

The PCR testing process is conducted, and the thermal profile is provided in Supplementary Fig. S7, including the reverse transcription time of 20 min and the thermocycling time (95 and 60 °C) of 60 min, with one thermocycle for 90 s (Supplementary Movie S7). It takes less time for FAST-POCT to complete one thermocycle (90 s) than that of conventional RT-PCR (180 s for one thermocycle). This can be attributed to the high surface-to-volume ratio and the small thermal inertia of the microscale PCR reaction chamber. The chamber surface is 96.6 mm² and the volume of the chamber is 25 mm³, making the surface-to-volume ratio about 3.86. It can be seen in Supplementary Figure S10 that there is a groove on the back of the PCR testing area of our platform, making the bottom thickness of the PCR chamber 200 μm. A heat-transferring elastic pad is adhered to the heating surface of the temperature control unit, ensuring the tight contact with the back surface of the testing chamber. In this way, the thermal inertia of the platform can be reduced, and the heating/cooling efficiency is increased. During the thermocycling process, the paraffin wax embedded in the platform melted and flowed into the PCR reaction chamber, acting as a sealing substance to prevent reagent evaporation and environment contamination (seen in Supplementary Movie S8).

All the above-mentioned PCR testing processes were fully automated using a customized FAST-POCT instrument, which consists of a programmed pressure controlling unit, magnetic extraction unit, temperature control unit, and fluorescent signal capture and processing unit. It should be noted that we used the FAST-POCT platform for RNA extraction and then used the extracted RNA samples to run PCR reactions using FAST-POCT system and the benchtop PCR system for comparison. The results are almost identical as shown in Supplementary Fig. S8. The operator performs the simple task of injecting the sample into the M chamber and inserting the platform onto the instrument. The quantitative testing results are then available after about 82 min. Detailed information about the FAST-POCT instrument can be found in Supplementary Figs. S9, S10 and S11.

### Characterization of the FAST-POCT platform for PCR testing of Influenza A and B virus

Influenza caused by influenza virus A (IAV), B (IBV), C (ICV), and D (IDV) is a common global phenomenon. Among these, IAV and IBV are responsible for most cases of severe illness as well as seasonal epidemics, which infect 5–15% of the global population causing 3–5 million cases of severe illness and accounting for 290,000–650,000 deaths each year due to respiratory illness[56,57]. Early diagnosis of IAV and IBV is critical to reducing morbidity and its associated economic burdens. Among available diagnostic techniques, the reverse transcriptase polymerase chain reaction (RT-PCR) is considered the most sensitive, specific, and accurate (>99%)[58,59]. However, conventional RT-PCR techniques require several pipetting, mixing, metering, and liquid transferring operations, limiting access to professional personnel in resource-limited settings. Here, the FAST-POCT platform was applied for PCR testing of IAV and IBV separately to obtain their lower limit of detection (LOD). Additionally, multiplexed testing of IAV and IBV was conducted to differentiate different pathotypes of a species, providing a promising genetic analysis platform and an opportunity for accurate treatment to diseases.

Figure 5a shows the PCR testing results for IAV using 150 μl purified viral RNA as samples. Figure 5a(i) shows that when the concentration of IAV was $10^6$ copies/ml, the fluorescence intensity (ΔRn)

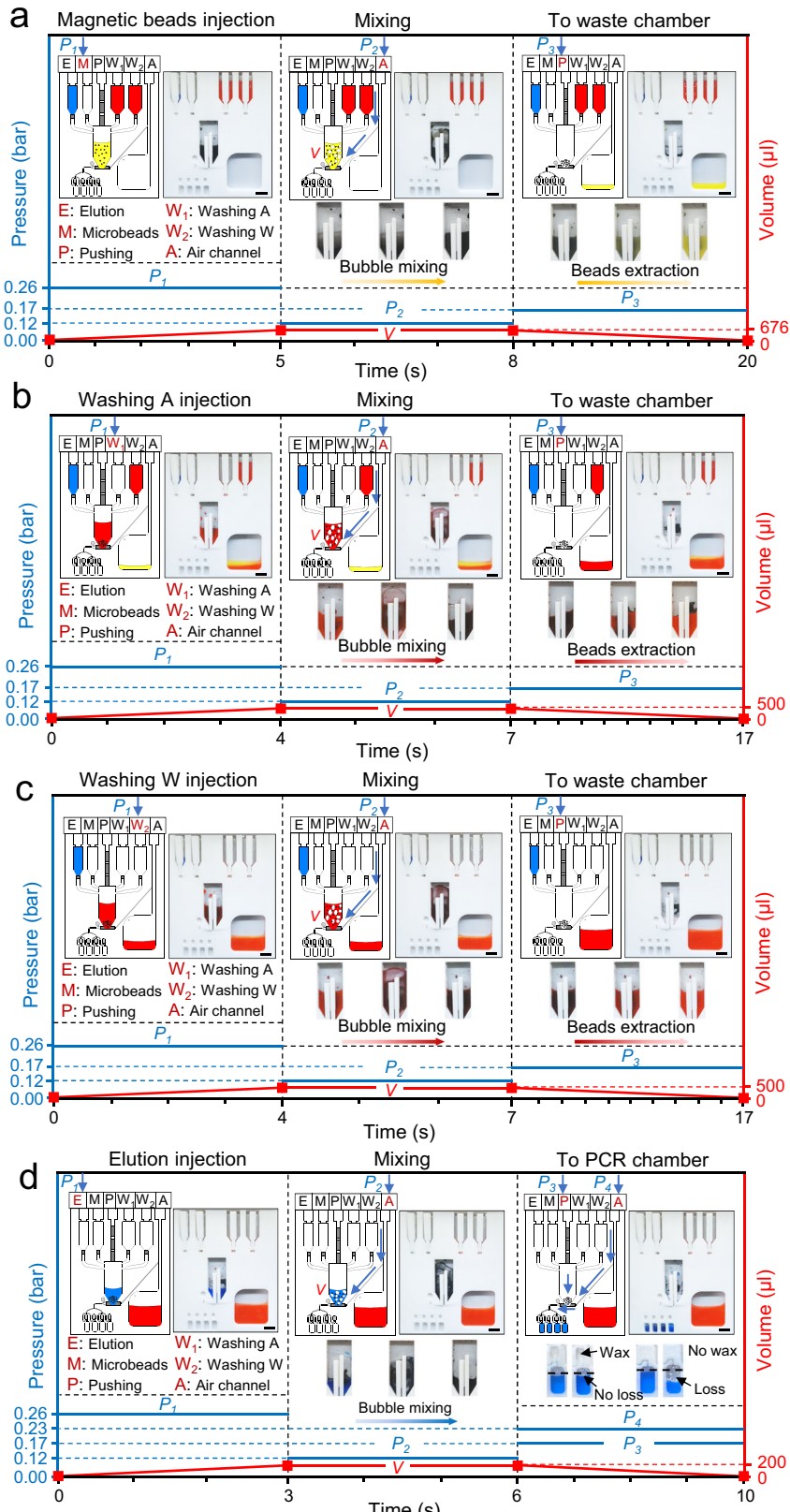

**Fig. 4 | Illustration for working principle of FAST-POCT platform. a** RNA adsorption step, the sample is introduced into the M inlet and injected into the mixing chamber together with the pre-stored microbeads solution. After mixing and beads extraction, the reagent is dispensed to the waste chamber. **b, c** Washing step, different pre-stored washing reagents are injected into the mixing chamber and after mixing and beads extraction, the reagents are transferred to the waste chamber. **d** Elution step, the elution reagent is injected, and after mixing and beads extraction, the reagent is transferred to the PCR reaction chamber. The curves show the working process and the related parameters in different steps. The pressure is the applied pressure through various chambers. The volume is the volume of the liquid in the mixing chamber. All the scale bars are 1 cm. Source data are provided as a Source Data file.

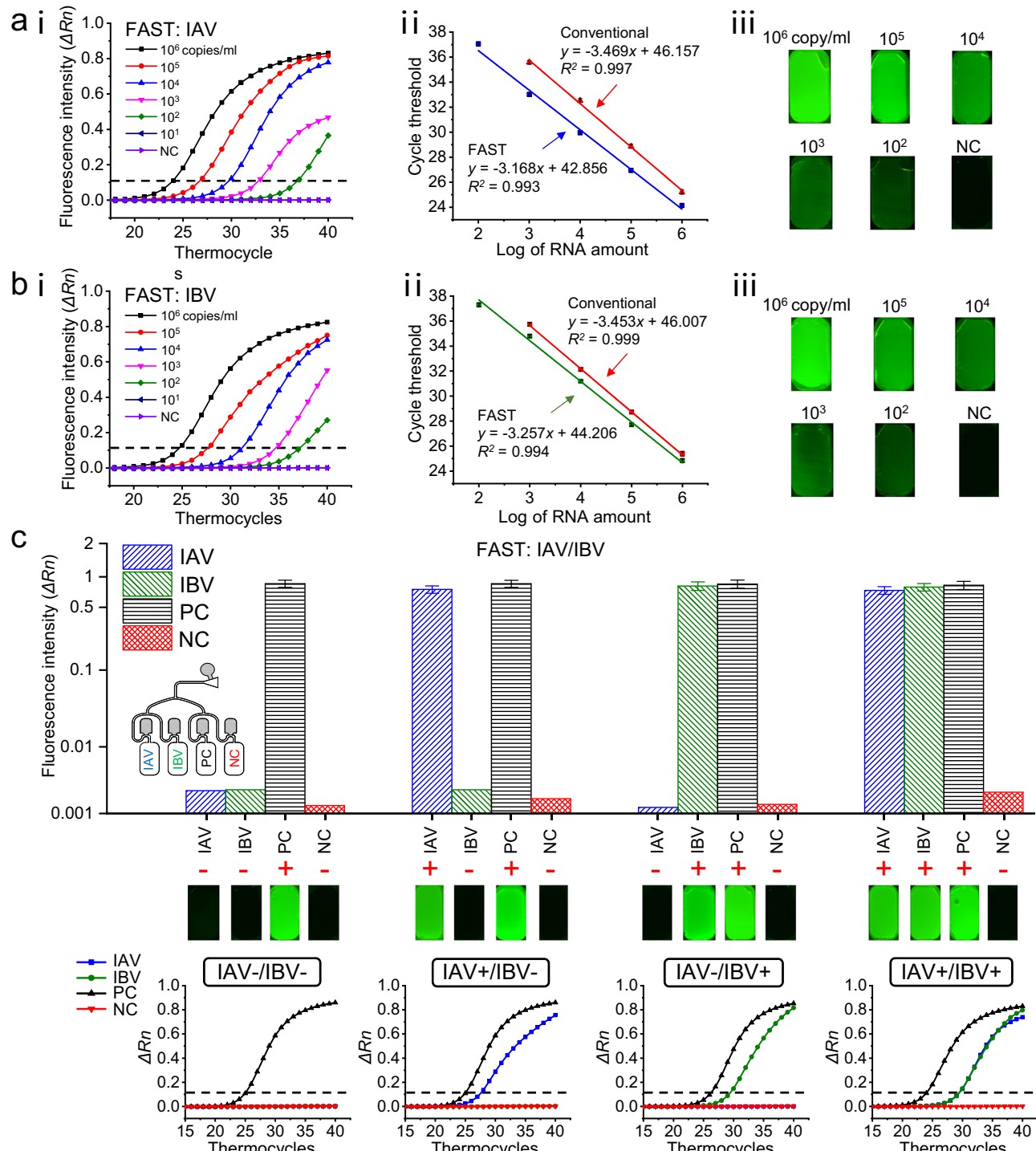

**Fig. 5 | PCR testing results for influenza virus. a** PCR testing results for influenza A virus (IAV) with IAV concentration ranging from $10^6$ to $10^1$ copies/ml using TE buffer as the negative control (NC). (i) Real-time fluorescence profile. (ii) The linear calibration curve between the log concentration of IAV RNA and the cycle threshold (Ct) for both FAST and conventional testing technique. (iii) Fluorescence images of FAST-POCT with IAV after 40 cycles. **b**, PCR testing results for influenza B virus (IBV) with (i) real-time fluorescence profile. (ii) the linear calibration curve and (iii) fluorescence images of FAST-POCT with IBV after 40 cycles. The lower limit of detection (LOD) for IAV and IBV using FAST-POCT platform is $10^2$ copies/ml, which is lower than that of the conventional methods ($10^3$ copies/

ml). **c** Multiple detection results for IAV and IBV. GAPDH was used for positive control, and the TE buffer was used for negative control to prevent possible contamination and background amplification. Four different types of samples can be identified: (1) negative sample ("IAV-/IBV-") with only GAPDH; (2) IAV-infected ("IAV+/IBV−") with IAV and GAPDH; (3) IBV-infected ("IAV-/IBV+") with IBV and GAPDH; (4) IAV/IBV-infected ("IAV+/IBV+") with IAV, IBV, and GAPDH. The dotted lines denote the threshold line. $n$ = 6 biologically independent experiments were conducted with the data shown as ± s.d. Source data are provided as a Source Data file.

can be 0.830, and as the concentration decreased to $10^2$ copies/ml, $\Delta Rn$ can still be as high as 0.365, which is about 100-fold higher than that of the blank negative control group (0.002). For quantitative analysis, a linear calibration curve between the log concentration of IAV and the cycle threshold (Ct) was obtained (Fig. 5a(ii)) with $R^2 = 0.993$ in the range of $10^2$–$10^6$ copies/ml based on six separate experiments. These results agree well with the conventional RT-PCR techniques. Figure 5a(iii) shows the fluorescence images of the testing results after 40 cycles from FAST-POCT platform. We found that the FAST-POCT platform can detect as low as $10^2$ copies/ml IAV. However, the conventional method does not have a $Ct$ value in the concentration of $10^2$ copies/ml, making its LOD about $10^3$ copies/ml. We assume this may be attributed to the high efficiency of the bubble mixing. The PCR testing experiments for purified IAV RNAs were conducted to evaluate different mixing methods, including shaking mixing (the same mixing method as the conventional RT-PCR operation), bubble mixing (the present method, 3 s at the pressure of 0.12 bar) and without mixing as a control group. The results can be found in Supplementary Fig. S12. It can be seen when the RNA concentration is high ($10^6$ copy/ml), the $Ct$ value for different mixing methods is almost the same with the $Ct$ value of bubble mixing. As the RNA concentration decreases to $10^2$ copy/ml, the shaking mixing and the control group exhibit no $Ct$ value while the bubble mixing method still obtains a $Ct$ value of 36.9, which is below the threshold Ct value of 38. The results show the advantage of bubble mixing, which is also demonstrated in other literature[60] and it may also explain the reason why the FAST-POCT platform has a slightly higher sensitivity than that of the conventional RT-PCR. Figure 5b shows the PCR testing results for purified IBV RNA samples, with the concentration ranging from $10^1$ to $10^6$ copies/ml. The results are similar to the IAV testing, which achieved $R^2 = 0.994$ and LOD of $10^2$ copies/ml.

Figure 5c shows the multiplexed testing results for IAV/IBV. Here, the viral lysates were used as the sample solution instead of the purified RNAs, with four primers targeting IAV, IBV, GAPDH (positive control), and TE buffer (negative control) added to the four different reaction chambers of the FAST-POCT platform. The positive and negative control used here is to prevent possible contamination and background amplification. The testing was classified into four groups: (1) negative sample ("IAV−/IBV−") with only GAPDH; (2) IAV-infected ("IAV+/IBV−") with IAV and GAPDH; (3) IBV-infected ("IAV−/IBV+") with IBV and GAPDH; (4) IAV/IBV-infected ("IAV+/IBV+") with IAV, IBV, and GAPDH. Figure 5c shows that when a negative sample was applied, the positive control chamber exhibited fluorescence intensity $\Delta Rn$ of 0.860 with $\Delta Rn$ of IAV and IBV similar to that of negative control (0.002). For IAV+/IBV−, IAV−/IBV+ and IAV+/IBV+ groups, the IAV/GAPDH, IBV/GAPDH, and IAV/IBV/GAPDH chambers presented significant fluorescence intensity, respectively, with other chambers showing florescence intensities at the background level even after 40 thermocycles. From the above tests, the FAST-POCT platform shows prominent specificity and allows us to pathotype different influenza viruses at once.

### POCT application with clinical samples collected on site
To validate FAST-POCT's clinical applicability, we tested 36 clinical samples (nasal swab samples) from patients ($n = 18$) with IBV and control individuals ($n = 18$) without IBV (Fig. 6a). Patient information is available in Supplementary Table 3. IBV infection status was independently confirmed and the study protocols were approved by The First Affiliated Hospital of Zhejiang University (Hangzhou, Zhejiang). Each patient sample was divided into two categories. One aliquot was processed using the FAST-POCT and the other using a benchtop PCR system (SLAN-96P, China). Both assays used the same purification and testing kits. Figure 6b shows the results from FAST-POCT and conventional PCR with reverse transcription (RT-PCR). We compared fluorescence intensity (FAST-POCT) with $-\log_2(Ct)$, where $Ct$ is the cycle cut-off of the conventional RT-PCR. A good concordance between these two methods was observed. FAST-POCT and RT-PCR

showed a strong positive correlation with the Pearson coefficient ($r$) values of 0.90 (Fig. 6b). We next assessed the diagnostic accuracy of FAST-POCT. As independent analytical measures, the fluorescence intensity (FL) distribution is provided for both positive and negative samples (Fig. 6c). The FL values were significantly higher (***$P = 3.31 \times 10^{-19}$; two-sided t-test) in patients with IBV than those in control groups (Fig. 6d). Receiver operating characteristic (ROC) curves were further constructed for IBV. We found the diagnostic accuracy was excellent, with an area under the curve of 1 (Fig. 6e). Please note that because of the mandatory mask order in China since 2020 due to COVID-19, we did not find patients with IAV; as a result, all the positive clinical samples (i.e., nasal swab samples) are only for IBV.

## Discussions
In this paper, we demonstrated a FAST, which possesses the desired features for an ideal POCT. The advantages of our technology include: (1) multifunctional dispensing (in cascaded, simultaneous, sequential, and selective manner), on-demand releasing (rapid and proportional releasing to the applied pressure) and robust operation (without leakage under the vibration of 150 rad/min); (2) long-term storage (accelerated life tests of 2 years with about 0.3% weight loss); (3) capability in handling the liquid with a wide range of wettability and viscosity (viscosity as high as 5500 cp); (4) cost-effectiveness (estimated material costs for the FAST-POCT PCR device is about $1). By combining the multifunctional dispensing units, an integrated FAST-POCT platform was demonstrated and applied to PCR testing of influenza A and B viruses. The IAV and IBV can be detected in situ with $10^2$ copies/ml of LOD, and the one-step pathotyping of IAV and IBV on the FAST-POCT platform was achieved in 82 min. The clinical tests with 36 nasal swab samples showed good concordance in fluorescence intensity with standard RT-PCR (Pearson coefficients > 0.9). Parallel to this work, varieties of emerging biochemical techniques (i.e., plasmonic thermocycle testing, amplification-free immunoassay, and nanobody-functionalized testing) have shown their potential for POCT. However, due to a lack of an all-integrated and robust POCT platform, these techniques inevitably require separate pre-treatment processes (e.g., RNA extraction[44], incubating[45], and rinsing[46]), which further makes the present work complementary to these technologies to achieve advanced POCT capabilities with desired "sample-in-answer-out" performance. In this work, though a pneumatic pump that is used to activate the FAST valves is small in size and can be integrated into a benchtop instrument (Figs. S9, S10), it still consumes considerable power and makes noises. In principle, the pneumatic pump can be replaced by other means for a smaller form factor, such as use of an electromagnetic force or finger-actuated forces. Further improvements can include, for example, customizing the cartridge for different and specific biochemical assays, and adopting novel detection method with no need for the heating/cooling system, leading to an instrument-free POCT platform for PCR applications. We believe that the proposed FAST technique represents a potential to establish a universal platform not just for biomedical testing, but also for environmental monitoring, food quality inspection, material synthesis, and pharmaceuticals, given that the FAST platform provides a means to manipulate fluids.

## Methods
### Study participants
The collection and use of human nasal swab samples were approved by the ethics committee of the First Affiliated Hospital of Zhejiang University (IIT20220330B). 36 nasal swab samples were collected, involving 16 adults < 30 years old, 7 adults > 40 years old, and 19 males, 17 females. The demographics are provided in the Supplementary Table 3. Informed consent was obtained from all the participants. The participants are all influenza suspected people and volunteered to be tested with no compensation.

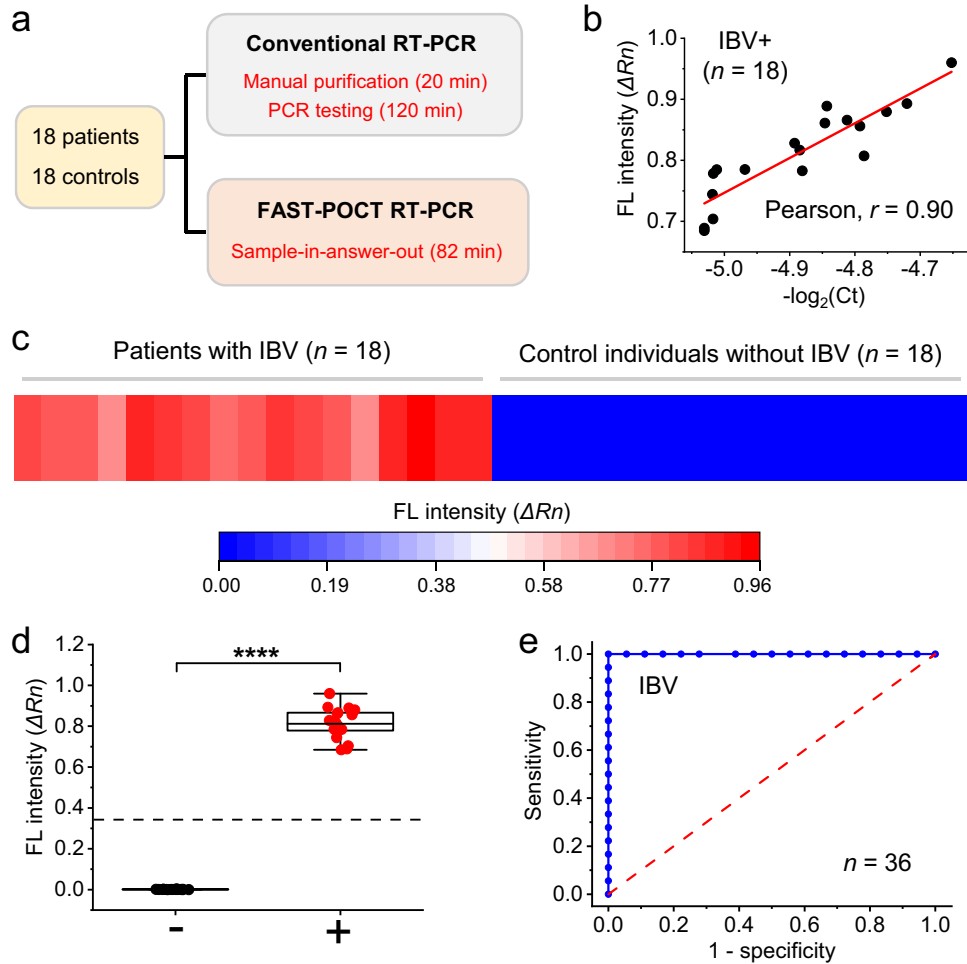

**Fig. 6 | Testing of clinical samples with FAST-POCT platform for Influenza diagnosis. a** Clinical study design. A total of 36 samples were analyzed by FAST-POCT platform and conventional RT−PCR, including 18 patients' samples and 18 control individuals without Influenza. **b** Evaluation of analytical concordance between FAST-POCT PCR and conventional RT-PCR. The results were positively correlated (Pearson's $r = 0.90$). **c** Fluorescence intensity levels for 18 patients with IBV and 18 controls. **d** FL values were significantly higher in the patients with IBV (+) than in controls (−) (\*\*\*\*$P = 3.31 \times 10^{-19}$; two-sided t-test; $n = 36$). For each box plot, the central black mark indicates the median, and the bottom and top lines of the box indicates the 25th and 75th percentiles, respectively. The whiskers extend to the min-max data points that were not considered outliers. **e** ROC curves. The dotted line **d** denotes the cutoff estimated from the ROC analysis. The AUC was 1 for IBV. Source data are provided as a Source Data file.

## Fabrication

The substrate and cover of FAST were made of polylactic acid (PLA) materials and was 3D printed using an Ender 3 Pro 3D printer (Shenzhen Creality 3D Technology Co. Ltd.). The double-sided adhesive was purchased from Adhesives Research, Inc., and the mode is 90880. The 100 μm-thick PET film was purchased from McMaster-Carr. The adhesives and PET films were both cut using a Silhouette Cameo 2 cutter from Silhouette America, Inc. The elastic film was made of PDMS material using a mold casting method. First, a PET frame with thickness of 200 μm was cut using a laser system and adhered to a 3 mm thick PMMA sheet using double-sided tapes with thickness of 100 μm. Then, a PDMS precursor (Sylgard 184; part A:part B = 10:1, Dow Corning) was cast to the mold and excessive PDMS was removed using a glass rod. When subjected to 3 h of curing at 70 °C, the PDMS film with a thickness of 300 μm can be peeled off from the mold.

## Setups for FAST performance tests

The pictures of the multifunctional dispensing, on-demand release, and robust operations were all taken using a high-speed camera (Sony AX700 with 1000fps). The orbital shaker used in the robustness tests was purchased from SCILOGEX (SCI-O180). An air compressor was used to generate the air pressure, and several digital precision

pressure regulators were used to adjust the value of the pressure. The flow behavior testing process is the following. A given amount of liquid was injected into the testing device and a high-speed camera was applied to record the flow behavior. The still images were then captured from the flow behavior video at fixed time points and the remaining area was calculated using software Image-Pro Plus and then multiplied by the depth of the chamber to calculate the volume. Details on the flow behavior testing system can be found in Supplementary Fig. S4.

## Bubble mixing test

50 μl microbeads and 100 μl DI water were injected into a bubble mixing device. The pictures of the mixing performance were taken by a high-speed camera every 0.1 s with the pressure varying from 0.1 bar, 0.15 bar, and 0.2 bar, respectively. The pixel information during mixing can be obtained from these pictures using a photo processing software (photoshop CS6). And the mixing efficiency can be achieved using the following equation[53].

$$M = 1 - \sqrt{\frac{1}{N}\sum_{i=1}^{N}\left(\frac{c_i - \bar{c}}{\bar{c}}\right)} \qquad (1)$$

where $M$ is the mixing efficiency, $N$ is the total number of sample pixel points, $c_i$ and $\bar{c}$ are normalized concentration and expected normalized concentration. Mixing efficiency ranges from 0 (0%, not mixing) to 1 (100%, full mixed). The results are shown in Supplementary Fig. S6.

## Preparation for influenza virus PCR tests

The IAV and IBV real time RT-PCR kit including the IAV and IBV RNA sample (catalog number: RR-0051-02/RR-0052-02, Liferiver, China), the Tris-EDTA buffer (TE buffer, catalog number: B541019, Sangon Biotech, China), the RNA purification kit (catalog number: Z-ME-0010, Liferiver, China), and the GAPDH solution (catalog number: M591101, Sangon Biotech, China) for positive control are all commercial. The RNA purification kit includes bonding buffer solution, washing A, washing W, elution solution, magnetic microbeads, and AcrylCarrier. The IAV and IBV real time RT-PCR kit includes IFVA nucleic acid fluorescence PCR testing mixture and RT-PCR enzyme. 6 µl AcrylCarrier and 20 µl magnetic microbeads were added to the 500 µl bonding buffer solution, which was shaken and subsequently the microbead solution was obtained. 21 ml ethanol was added to washing A and W, which was shaken, after which the washing A and washing W solutions were obtained, respectively. 18 µl IFVA nucleic acid fluorescence PCR testing mixture and 1 µl RT-PCR enzyme were then added to the 1 µl TE solution, which was shaken and centrifuged for several seconds, thus producing the 20 µl primers for both IAV and IBV.

## Conventional RT-PCR testing procedures

The following RNA purification process was followed: (1) RNA adsorption. 526 µl microbeads solution was pipetted to a 1.5 ml centrifuge tube, to which was added a 150 µl sample; the tube was then manually shaken up and down for 10 times. The 676 µl mixture was transferred to the affinity column where it was centrifuged for 60 s with a rotation speed of $1.88 \times 10^4$ g. The subsequent waste solution was then abandoned. (2) Washing step one. 500 µl washing solution A was added to the affinity column; centrifuging occurred for 40 s with the rotation speed of $1.88 \times 10^4$ g, and the waste solution was subsequently abandoned. This washing process was repeated twice. (3) Washing step two. 500 µl washing solution W was added to the affinity column, followed by centrifuging for 15 s with the rotation speed of $1.88 \times 10^4$ g and then abandoning the waste solution subsequently. This washing process was repeated twice. (4) Elution. 200 µl elution solution was added to the affinity column, followed by centrifuging for 2 min with the rotation speed of $1.88 \times 10^4$ g. (5) RT-PCR: Elution solution was introduced to 20 µl primer solution in the PCR tube; the tube was then placed into a real time PCR testing equipment (SLAN-96P) to perform the RT-PCR process. The entire testing process took approximately 140 min (RNA purification was 20 min, and PCR testing was 120 min).

## FAST-POCT platform configuration

526 µl microbeads solution, 1000 µl washing A solution, 1000 µl washing W solution, 200 µl elution solution, and 20 µl primer solutions were pre-added and stored in the chambers M, W1, W2, E, and PCR testing chamber after the platform assembles. Then, the 150 µl sample was pipetted to chamber M and the FAST-POCT platform was inserted into the testing instrument shown in Supplementary Fig. S9. After about 82 min, the testing results were obtained.

## Statistical analysis

All testing results are presented as means ± s.d. after repeating at least six times using separate FAST-POCT platforms with biologically independent samples unless otherwise specified. No data were excluded from the analyses. The experiments were not randomized. The investigator was not blinded to group allocation during the experiment.

## Reporting summary

Further information on research design is available in the Nature Research Reporting Summary linked to this article.

## Data availability

The data that support the findings of this study are available from Supplementary Information. Source data are provided with this paper.

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

## Acknowledgements

C.L. and H.J. acknowledge support from Vantronics LLC, Dr. Jun Fan from the First Affiliated Hospital of Zhejiang University (Hangzhou, Zhejiang) for the sample collection and independent verification. Some of the experimental work was conducted using the facilities at Arizona State University when C.L. and H.J. were at that institution. H.J. acknowledges support from Westlake University.

## Author contributions

C.L. and H.J. designed the experiments. C.L. and Z.Y. carried out experiments and analysis. C.L. and H.J. performed theoretical analysis and wrote the paper.

## Competing interests

The authors declare no competing interests.
