## [Peer Review File · Nature Communications]

A film-lever actuated switch technology for multifunctional, on-demand and robust manipulation of liquidsEditorial Note: Parts of this Peer Review File have been redacted as indicated to maintain individual's confidentiality.

REVIEWER COMMENTS

Reviewer #1 (Remarks to the Author):

In this manuscript by Liang and colleagues, the authors introduce film-lever actuated switch technology (FAST) that can manipulate liquids and facilitate the development of point-of-care (POC) diagnostic testing platforms. Every FAST unit uses a block, an elastic film, and a lever to form a normally-closed valve that prevents liquid flow. When a pressure is applied, however, the elastic film stretches and pushes the lever, which opens a gap above the block to allow liquid flow. Using FAST, the authors demonstrate multifunctional and on-demand dispensing, robust operations (e.g., no liquid leaking under vibration), and capability for long-term reagent storage (e.g., no liquid evaporation when sealed). The authors also constructed a FAST-based polymerase chain reaction (PCR) system that integrates reagent introduction, mixing and reaction functions all in one automated process. For use case demonstration, the authors detected Influenza A and B viral RNA with solid sensitivity (100 copies/mL) and no cross-reactivity between the two targets. Based on these results, the authors believe that their FAST-POCT platform promises robust automation of biomedical analysis, and thus can accelerate the commercialization of a range of POCT devices.

The strengths of this work include a new technology for manipulating liquids for the development of POC diagnostic testing platforms and comprehensive characterization of this new technology. The reviewer can agree with the authors that FAST presents an intriguing addition to existing technologies and the reviewer appreciates the characterization results, which sufficiently demonstrate the working principle of FAST and its liquid manipulation capability. On the other hand, the reviewer believe that the current work still must be significantly improved, especially in presenting the state of the art of POC diagnostic testing platforms, strengthening the data set on assay performance, and providing clarity and experimental details. The areas that must be improved are summarized in the comments below.

Major Comments

1. The authors must improve the introduction and properly introduce the state of the art of POC diagnostic testing platforms. Currently, the authors only focused on lab-on-a-disc technology. However, in the research space, there are many other POC-amenable technologies by various researchers (e.g., Gwo-Bin Lee of Taiwan, Changchun Liu of UConn, Haim Bau of UPenn, Jacqueline Linnes of Purdue, Tza-Huei Wang of Johns Hopkins). In the commercial space, Cepheid, Binx, and Visby all manufacture and commercialize POC-amenable cartridges and instruments. Emerging startup such as Talis Biomedical and Minute Molecular Diagnostics also have similar capabilities. The authors must discuss these research and commercial advances and properly acknowledge the state of the art. Otherwise, the claims can be misleading. For example, in lines 117 – 118, the authors stated that “The FAST-POCT platform outperforms most existing POCT devices, which are currently capable of only one-directional dispensing, even with the aid of additional actuation forces²⁸⁻³⁶.” But references 28 – 36 only encompass lab-on-a-disc technology and centrifugal microfluidic devices.

2. The authors must better describe the advantages of this lever design over existing normally-closed membrane valve-based POC-amenable platforms. The reviewer understands that FAST is a different (and therefore new) method, but does it really provide new capabilities over existing normally-closed membrane valve-based POC-amenable platforms (e.g., Gwo-Bin Lee’s body of work), which can perform similar liquid manipulation. Furthermore, one potential drawback to FAST is the large size of the cartridge, because long levers are often needed. The authors should provide the exact dimension of their various cartridges (the smaller cartridges for characterization data shown in Fig. 3 and Fig. S3 and the large FAST-POC cartridge shown in Fig. 4), especially because the scale bars provided in the current manuscript were imprecise. For example, in Fig. 1, the size of the US nickel is actually not 1 cm (as suggested by the scale bar). In Fig. S5b, the cartridge appears to be bigger than a 96-well PCR tube rack. In all images where the scale bars are still needed, the authors must improve the precision of their scale bars.

3. The authors must more clearly explain the design and the construction of the companion

instrument. The image of the instrument (Fig. S5C) is unfortunately insufficient. For example, it is unclear how the pressure is applied and regulated, how the heater is oriented (and what it looks like), how the electromagnet is placed (and what it looks like). Based on Fig. S5B, the authors showed that six FAST cartridges could be placed in the instrument, but such a placement likely necessitated six heaters and six electromagnets, as well as additional tubings and valves for pressure control. It is also difficult to picture how the authors managed to create sufficient contact with the heater to ensure properly thermocycling. Finally, if one heater is used to heat all six cartridges, the heater must be large, which would render "rapid" thermocycling nearly impossible.

4. The authors must strengthen the assay results beyond what Fig. 5 presented. The data presentation in Fig. 5 is redundant, and when the redundant plots are removed, the dataset becomes somewhat thin. For example, for Fig. 5a and 5b, (ii) and (iii) are essentially the same data set plotted as scatter plots and bar plots. For Fig. 5c, though bars and PCR curves are not identical, streamlined data presentation would only require one set of the plots instead of both. The authors should at least consider testing some contrived or clinical swab samples.

5. The authors should also provide additional results and information to support existing PCR results. In the current manuscript, because the authors used a custom instrument to perform PCR, more details and additional characterization results should be provided. For example, the fluorescence intensities shown in Fig. 5a and Fig. 5b are measured by the detector in the instrument, but what is the detector? Moreover, it would be helpful to visually observe increased fluorescence intensities at the end of PCR. Here, because the PCR chambers have open windows, it is possible to provide fluorescent images before and after PCR. The PCR condition – such as the thermocycling protocol and the primer concentration – should also be provided in much more detail. The reviewer also wondered why there did not seem to be a reverse transcription step in the thermocycling protocol. Based on the results shown in Fig. 5a(ii) and Fig. 5b(ii), the PCR efficiencies can be interpreted as greater than 100%. The authors should share their thoughts about the unexpectedly good efficiencies. Finally, the authors should also briefly mention how they loaded the PCR reagents. Did they inject the PCR reagents before they assemble the cartridge?

6. The authors should reconsider their claim that FAST-POCT "achieves 10-fold higher sensitive [sic] than the conventional PCR systems". The authors used different RNA extraction methods for their POC platform and their conventional PCR systems; making a direct comparison between methods with at least two variables (i.e., different extraction and different platform) seems non-ideal. At the very least, the authors must be clear about the differences. The authors may also want to at least compare the extraction method on benchtop and/or just PCR (sans RNA extraction) in FAST-POC versus benchtop instrument so that the improvement from FAST-POC can be better assessed.

7. The reviewer would like the authors to improve their reasoning for their observed improvement in FAST-POC. In lines 235 – 236, the authors presume that the improvement "may be due to the small thermal inertia and the high surface-to-volume ratio of the microscale PCR reaction chamber". The reviewer cannot agree with this reason. Based on the reviewer's estimate from Fig. S5, the PCR chamber area (the part that is visible to eye) appears comparable to 0.2 mL PCR tubes. The reaction volume in the PCR chamber was 20 μ L, which is on par with typical benchtop PCR. From what the reviewer can see, one possible way that the thermal mass of can be smaller for the PCR in FAST-POC is if the PCR is heated through the thin plastic film rather than the thicker substrate layer – a point that the authors unfortunately did not clarify in the current manuscript. As for the surface area to volume ratio, the reviewer can agree that it is likely slightly higher for the PCR chamber in FAST-POC, but it is also not more than an order of magnitude different, so it is difficult to the reviewer to believe that difference led to more efficient thermocycling.

8. The authors must better substantiate or contextualize the one-hour claim. First, the authors must provide detailed thermocycling protocol and temperature ramping performances of their instrument. Again, based on the layout of the instrument and the reviewer's experience, it seems really difficult to perform rapid thermocycling, even if a smaller heater that only covers the 4 PCR chambers in FAST-POC is used. Also, based on Fig. S8, the authors seem to be able to perform 18 cycles between 95 degrees C and 60 degrees C in ~40 min. The authors seem to perform 10-minute hot start at 95 degrees C, which is a rather lengthy. The authors also seem to have

skipped the reverse transcription step. If the authors used a really high concentration of target so that they can perform few PCR cycles so that the total turnaround time is under 60 min, that is OK, but the authors still must clearly state so.

9. The authors should also improve their discussion section. For example, the authors should discuss areas of improvement for their FAST technology. The reviewer can see that the authors may want to discuss ways to decrease the footprint of their cartridge (and hence their instrument). The authors can also see that the authors may want to discuss how to free their system of pumps and gauges.

Minor Comments

1. The reviewer thinks that mixing can also be presented as a stand-alone functionality of FAST. In the current manuscript, mixing was only included as part of Fig. 4, which appeared rather suddenly. The authors can consider including a mixing demonstration in Fig. 3 so that future readers can be more oriented for Fig. 4. The authors should consider making this change.

2. The authors should include one or two sentences in the caption of Fig. 2 to describe terminologies in Fig. 2c (i.e., B, q, Fs, Tx). The reviewer saw that these terms were defined in the text, but it would make digesting the information easier if these terms were also defined and contextualized in the caption.

3. The authors should improve the clarity of "T-shaped clip structure", which were mentioned in lines 92 and 96. The authors have many options here. They can add one or two descriptive sentences, label either Fig. 1 or Fig. 2, refer to Fig. S1, or do a combination of these. This said, the reviewer thinks it is unnecessary to describe the clip structure for assembling the cartridge. So, the authors can also remove the information from the main text and only include it in the supplementary information.

4. The authors should provide more experimental details to the "Setup for FAST performance tests" section in Methods. For example, how was the liquid volume in each chamber calculated and then presented in Fig. 3? Did the authors capture a video using their high-speed camera (should provide vendor information), take still images at fixed time points (using what software), calculate the area and then multiply by the depth of the chamber?

Reviewer #2 (Remarks to the Author):

The paper presents a new switching mechanism for liquid handling, in particular for point of care applications. The authors demonstrate a sample-in-answer-out PCR based analysis using their switch technology. The paper is well written and the structure is clear.

The switch provides an interesting technological approach for liquid handling. From my perspective, the impact and the scientific value of the presented work, however, falls short for a publication in Nature Communication. Point-of-Care nucleic acid analysis has been discussed in numerous papers and there are products on the market with the most famous one being probably the Cepheid GeneXpert system.

As the authors mention, centrifugal microfluidics is one of the most promising candidates. Here, also products for sample in answer out testing are on the market:

Meridianbioscience with the revogene:

<https://www.meridianbioscience.com/diagnostics/platforms/molecular/revogene/?country=US>

Spindiag with the Rhonda system:

<https://www.spindiag.de/spindiag-rhonda-pcr-rapid-covid-19-test-watch/>

The authors claim that forces directing radially outwards is a significant challenge in centrifugal microfluidics but this challenge has been solved. Clime et al. *Lab Chip*, 2015, 15, 2400-2411 discusses an active approach for liquid control. Zehnle et al. presented a fully passive approach (*Lab Chip*, 2012, 12, 5142-5145). The passive approaches with no extra means have been applied by Czilwik et al. (*Lab Chip*, 2015, 15, 3749-3759) or Rombach et al. (*Analyst*, 2020, 145, 7040-

7047) for sample-in answer out of nucleic acid analysis with pre-stored reagents.

Taking these publications into considerations, even though new, I do not see sufficient impact for the presented technology and not enough novelty for the discussed application that justifies publication in Nature Communications. I do think the technology is valuable and in summary worth to be published. I thus recommend that the authors select a journal that is more specific to the topic.

A more specific comment:

The authors claim that the presented RT-PCR is more sensitive than a conventional RT-PCR system. I could not find a reasonable explanation and in particular not a discussion on why this increase of sensitivity could be related to their technology, thus, such a statement is misleading from my perspective. The claim: "This may be due to the small thermal inertia and the high surface-to-volume ratio of the microscale PCR reaction chamber, as well as the high efficiency of the bubble mixing" is highly speculative. For example, if you compare bubble mixing with the used mixing by hand, this may be the case, then, why did the authors not use Vortex mixing? I would assume that if one optimizes the assay procedure for the reference system, a similar sensitivity would be reached.

Reviewer #3 (Remarks to the Author):

The manuscript by Liang and colleagues describes a mechanical valving technique that is used to control liquids flow in microfluidic devices. The mechanism provides an incremental improvement compared with similar techniques found in the literature. The introduction is rather superficial, and the claims are exaggerated.

While the functionality of the method is explained well in some cases the authors do not explain the source of the pneumatic pressure. A successful device should not require expensive, bulky equipment, on contrary it has to be operated without any external source to apply pressure or actuate the system. The functionality of the device should be investigated with liquids with different viscosities and mechanical properties and under various climate conditions. The details of how the external pressure is applied are not presented. The dependency on the use of external pressure for opening and closing does not make the POCT an ideal solution for the POCT test.

The results are compared with existing liquid handling techniques, this makes it impossible for the reader to see the advantage/disadvantage of their technique contrast especially when compared with the liquid handling methods that are already commercialized e.g., cobas® Liat®. The authors claim that film-lever actuated switch technology provides full control in liquid handling. However, regarding the materials used, fabrication technique, and complexity of their design, the repeatability, and the robustness of their method necessitates more experiments with different liquids and in different environments. The POCT must be able to perform in extreme climate conditions and therefore the 0.8% liquids loss 60 days at room temperature is remarkably high. This is a common drawback of storing liquids directly in a diagnostic platform, which is why aluminum, glass, and micro-dispenser containers are applied.

The technology developed in this study is potentially interesting for POCT, however, there are several limitations. The authors claimed the medical relevance of their device, but they miss the point of diagnostic POCT. There is a clear misalignment between the technology development and the actual diagnostic capability of their device. To be successful a diagnostic technology should complete the cycle from "sampling to treatment", but the paper lacks the whole treatment, usability, and user-interaction of their potential POCT.

- Justifying the diagnostic-amenability by a proof-of-concept PCR test for influenza virus is not enough for justifying the POCT potential. I suggest the author resize their medical relevant claims.

- They fail to cite the right diagnostic literature. Citations 1,2,3,4 should be complemented with the right relevant report from e.g., the WHO (when cited) or analyses of diagnostic potentials from a health system relevance (e.g., Madhukar Pai research work). They re-cite statements from other

papers, without actually providing the right original reference.

After implementing correction, the article would be more suitable for a technical journal, as it is a proof-of-concept type of study. Such liquid handling technology has potential relevance, however claiming its diagnostic potential is too preliminary at the stage the device is.

For the first reviewer:

In this manuscript by Liang and colleagues, the authors introduce film-lever actuated switch technology (FAST) that can manipulate liquids and facilitate the development of point-of-care (POC) diagnostic testing platforms. Every FAST unit uses a block, an elastic film, and a lever to form a normally-closed valve that prevents liquid flow. When a pressure is applied, however, the elastic film stretches and pushes the lever, which opens a gap above the block to allow liquid flow. Using FAST, the authors demonstrate multifunctional and on-demand dispensing, robust operations (e.g., no liquid leaking under vibration), and capability for long-term reagent storage (e.g., no liquid evaporation when sealed). The authors also constructed a FAST-based polymerase chain reaction (PCR) system that integrates reagent introduction, mixing and reaction functions all in one automated process. For use case demonstration, the authors detected Influenza A and B viral RNA with solid sensitivity (100 copies/mL) and no cross-reactivity between the two targets. Based on these results, the authors believe that their FAST-POCT platform promises robust automation of biomedical analysis, and thus can accelerate the commercialization of a range of POCT devices.

The strengths of this work include a new technology for manipulating liquids for the development of POC diagnostic testing platforms and comprehensive characterization of this new technology. The reviewer can agree with the authors that FAST presents an intriguing addition to existing technologies and the reviewer appreciates the characterization results, which sufficiently demonstrate the working principle of FAST and its liquid manipulation capability. On the other hand, the reviewer believe that the current work still must be significantly improved, especially in presenting the state of the art of POC diagnostic testing platforms, strengthening the data set on assay performance, and providing clarity and experimental details. The areas that must be improved are summarized in the comments below.

Thanks for the encouragement and very detailed and constructive comments, which motivates us to significantly improve the quality of this work.

Major Comments

1. The authors must improve the introduction and properly introduce the state of the art of POC diagnostic testing platforms. Currently, the authors only focused on lab-on-a-disc technology. However, in the research space, there are many other POC-amenable technologies by various researchers (e.g., Gwo-Bin Lee of Taiwan, Changchun Liu of UConn, Haim Bau of UPenn, Jacqueline Linnes of Purdue, Tza-Huei Wang of Johns Hopkins). In the commercial space, Cepheid, Binx, and Visby all manufacture and commercialize POC-amenable cartridges and instruments. Emerging startup such as Talis Biomedical and Minute Molecular Diagnostics also have similar capabilities. The authors must discuss these research and commercial advances and properly acknowledge the state of the art. Otherwise, the claims can be misleading. For example, in lines 117 – 118, the authors stated that “The FAST-POCT platform outperforms most existing POCT devices, which are currently capable of only one-directional dispensing, even with the aid of additional actuation forces²⁸⁻³⁶.” But references 28 – 36 only encompass lab-on-a-disc technology and centrifugal microfluidic devices.

Appreciate your constructive comments. We have rewritten the introduction section to analyze advantages and disadvantages of existing POC technologies (e.g., centrifuge-dynamic method, active pneumatic method and pneumatic with normally closed valve method). We also made a comparison between the performance of the existing ones and our technology (Supplementary Table 2). To the best of our knowledge, in spite of the excellent contributions made by the previous researchers, to date there are still limited microfluidic technologies that *simultaneously* meet desired performance, including multifunctional dispensing, on-demand release, robust operations, long-term storage, high-viscosity liquid handling, and cost-effective fabrication, all of which currently are barriers for the adoption of POCT products (e.g., Cepheid, Binx, Visby, Cobas Liat and

Rhonda) in the open market.

The newly added text in the introduction section now reads:

Combining pneumatic operations in centrifugal microfluidics, e.g., the centrifugo-dynamic method^{37~39} and the active pneumatic method⁴⁰ have shown to be an appealing alternative. In the centrifugo-dynamic method, an additional cavity and connecting microchannels are integrated in the device to enable both outward and inward manipulations, though its pumping efficiency (ranging from 75% to 90%) depends highly on number of pumping cycles as well as the viscosity of the fluids. In the active pneumatic method, on-demand fluid releasing and inward manipulation are achieved by simultaneously applying positive pressure and precisely matched rotational speed, enabled by a high-speed motor. There have been other successful methods that employ only pneumatic driving mechanisms (positive^{41,42} or negative pressure⁴³) and a normally closed valve structure. By applying pressure sequentially in the pneumatic chamber, the liquid is pumped forward in a peristaltic way by which the normally closed valve avoids the liquid from flowing back, thereby enabling complex fluid manipulations. To the best of our knowledge, however, currently there are only limited microfluidic technologies that can perform complex fluid handling in a single POCT device, which includes multifunctional dispensing, on-demand release, robust operation, long-term storage, high-viscosity liquid handling, and cost-effective fabrication – all at the same time. A lack of multi-step functionality operating also could be one of the reasons why only a few commercial POCT products (e.g., Cepheid, Binx, Visby, Cobas Liat and Rhonda) to date have been successfully adopted in the open market.

Supplementary Table 2 compares the performance of existing POC technologies and FAST.

Supplementary Table 2. Comparisons among some POCT PCR studies and technologies.

Detection performance						
Reference	FAST-POCT (this work)	Gzilwik, et al, Lab Chip, 15, 3749- 3759 (2015)	Clime, et al, Lab Chip, 15, 2400- 2411 (2015)	Wang, et al, Biosens. Bioelectron, 41, 484-491 (2013)	Rombach, et al, Analyst, 145, 7040- 7047 (2020)	Jung, et al, Biosens. Bioelectron, 68, 218-224 (2015)
Target	Influenza RNA	Bacterial pathogens DNA	DNA	HIV DNA	Respiratory tract infection pathogens	Influenza A RNA
Method	RT-PCR	RT-PCR	Only DNA extraction	RT-PCR	RT-PCR	Isothermal PCR
Detection limit	15 copies	200 cfu of S. agalactiae	~1	62 copies	–	10 copies
Testing time	82 min	225 min	–	95 min	200 min	45 min
Fluid handling capability						
Multifunctional dispensing	Cascaded, simultaneous, sequential and selective	Inward dispensing	Inward dispensing	–	Inward dispensing	Two-way dispensing

	dispensing					
On-demand releasing	Rapid and proportional to the applied pressure	-	Yes	Yes	-	-
Robust operation	No leaking under the vibration of 150 rad/min	-	-	-	-	-
Long-term storage	Accelerated life tests of DI water with weight loss less than 0.3% for two years	-	-	-	Yes	-
Liquid properties	Manipulating the liquid with viscosity as high as 5,500 cP	Viscosity of 16 cP with efficiency of 75%	-	-	-	-
Driving mechanism	Positive pressure	Centrifugal force	Centrifugal force and positive pressure	Negative pressure	Centrifugal force	Centrifugal force

¹ The dash “-” means that the related results or features were not demonstrated in the paper.

2. The authors must better describe the advantages of this lever design over existing normally-closed membrane valve-based POC-amenable platforms. The reviewer understands that FAST is a different (and therefore new) method, but does it really provide new capabilities over existing normally-closed membrane valve-based POC-amenable platforms (e.g., Gwo-Bin Lee’s body of work), which can perform similar liquid manipulation. Furthermore, one potential drawback to FAST is the large size of the cartridge, because long levers are often needed. The authors should provide the exact dimension of their various cartridges (the smaller cartridges for characterization data shown in Fig. 3 and Fig. S3 and the large FAST-POC cartridge shown in Fig. 4), especially because the scale bars provided in the current manuscript were imprecise. For example, in Fig. 1, the size of the US nickel is actually not 1 cm (as suggested by the scale bar). In Fig. S5b, the cartridge appears to be bigger than a 96-well PCR tube rack. In all images where the scale bars are still needed, the authors must improve the precision of their scale bars.

Thanks for the comment. The existing normally closed membrane valve-based POC-amenable platforms by Gwo-Bin Lee’s group provide diverse means to handle liquid and contributes to the field significantly. In the platform developed by Gwo-Bin Lee’s group, by applying pressure sequentially in the pneumatic chamber, the liquid can be pumped forwards in a peristaltic way and the normally closed valve can avoid the liquid from flowing back with great potential for complex fluid manipulation; however, the authors have yet to address the long-term storage issue, which is critical for a successful POCT platform. Another likely advantage of the FAST method when compared with the normally closed membrane valve-based method is its readiness for mass production since it does not require high-precision alignment between the substrate and the elastic film layer, as opposed to the normally closed membrane valve-based method, which requires more precise alignment between the pneumatic layer and the liquid channel to ensure the precise manipulation of liquid. We

added the above discussion in the introduction section with related references.

The related text reads:

Some other successful methods with only pneumatic driving mechanisms (positive^{41,42} or negative pressure⁴³) and a normally closed valve structure have also been proposed. In these methods, pressure is applied sequentially in the pneumatic chamber, so that the liquid can be pumped forward in a peristaltic way, thereby allowing the normally closed valve to avoid the liquid from flowing back, and subsequently increasing the potential for complex fluid manipulation.

The dimension of the FAST-based cartridge varies depending on the specific applications. **Figure 2d** shows that the length of the lever can be adjusted from several millimeters to centimeters with the parameter t/l affecting the critical pressure. These parameters can be tuned to accommodate for desired liquid volume. To establish a fair comparison, the present POCT-FAST platform uses the same volume used in commercial purification and testing kits (i.e., two separate 1 milliliter wash solutions) for PCR experiments, and for this reason, the chamber needs to be large enough to contain the reagents. We provided the schematic diagram of all the devices showing their precise dimensions (**Fig. S2**) and updated the scale bar in the revised manuscript.

Figure S2. Schematic diagram for the dimensions of FAST-POCT in the unit of mm. a, FAST-POCT PCR device. b, Cascaded device. c, Simultaneous device. d, Sequential device. e, Selective device. f, On-demand testing device. g, Robustness testing device. h, Long-term storage testing device. The thickness of all the devices is 7 mm.

3. The authors must more clearly explain the design and the construction of the companion instrument. The image of the instrument (Fig. S5C) is unfortunately insufficient. For example, it is unclear how the pressure is

applied and regulated, how the heater is oriented (and what it looks like), how the electromagnet is placed (and what it looks like). Based on Fig. S5B, the authors showed that six FAST cartridges could be placed in the instrument, but such a placement likely necessitated six heaters and six electromagnets, as well as additional tubings and valves for pressure control. It is also difficult to picture how the authors managed to create sufficient contact with the heater to ensure properly thermocycling. Finally, if one heater is used to heat all six cartridges, the heater must be large, which would render “rapid” thermocycling nearly impossible.

Because of the nature of the proprietary information of the FAST-based POCT instrument, the original Fig. S5c (now Fig. S10) just shows an illustration of the prototype of the instrument. In order to address the reviewer’s comment without compromising the proprietary information, we demonstrated this instrument with only one FAST cartridge (Fig. S10) to showcase the necessary components of the instrument. As shown in Fig. S10, the dimension of one module is $L \times W \times H = 138 \times 109 \times 134$ mm and an instrument can have several modules of this dimension. On the back of the PCR testing area of the FAST cartridge is a groove that holds the temperature control unit (Fig. S10b), where a heat-transferring elastic pad is adhered to the heating surface, ensuring a tight contact with the back surface of the testing chamber. Figure S11 provides a detailed flow chart of the working principles of the FAST-POCT instrument. Figure S7 provides the thermocycling protocol, including the reverse transcription time of 20 min and the thermocycling time of 60 min, with one thermocycle for 90 s.

Figure S10. Illustration of the FAST-POCT testing module. a, The demonstration of the way to place the FAST-POCT device into a testing module prototype. b, i) The schematic diagram of the PCR testing unit. ii) Picture of the front view of the testing module prototype. iii) Picture of the back view of the testing module prototype. c, Flow chart of the relationship of different components. The scale bars are 10 mm.

Figure S7. Temperature profile of the testing process. a, Reverse transcription time is 20 min. Thermocycling time is 60 min. One thermocycling time is 90 s. b, Thermal images during PCR testing for one thermocycle, showing fairly stable thermal behavior.

4. The authors must strengthen the assay results beyond what Fig. 5 presented. The data presentation in Fig. 5 is redundant, and when the redundant plots are removed, the dataset becomes somewhat thin. For example, for Fig. 5a and 5b, (ii) and (iii) are essentially the same data set plotted as scatter plots and bar plots. For Fig. 5c, though bars and PCR curves are not identical, streamlined data presentation would only require one set of the plots instead of both. The authors should at least consider testing some contrived or clinical swab samples.

Appreciate this comment. In the revised manuscript, we deleted the redundant plots and provided the fluorescence images in **Fig. 5**. To validate FAST-POCT's clinical applicability, we tested 36 clinical samples (nasal swabs samples) from patients (n=18) with IBV and control individuals (n=18) without IBV (**Fig. 6a**). Patient information is provided in **Supplementary Table 3**. IBV infection status was independently confirmed by The First Affiliated Hospital of Zhejiang University (Hangzhou, Zhejiang). Dr. Jun Fan (a clinical physician) from this hospital was responsible for sample collection and independent verification. Each patient sample was divided into two. One aliquot was processed using the FAST-POCT and the other using a benchtop PCR system (SLAN-96P, China). Both assays used the same purification and testing kits. **Figure 6b** shows the results from FAST-POCT and conventional PCR with reverse transcription (RT-PCR). We compared raw fluorescence intensity (FAST-POCT) with $-\log_2(\text{Ct})$, where Ct is the cycle cut-off of the conventional RT-PCR. A good concordance between these two methods was observed. FAST-POCT and the conventional RT-PCR showed a strong positive correlation with the Pearson coefficient (r) values of 0.90 (**Fig. 6b**). We next assessed the diagnostic accuracy of FAST-POCT. As independent analytical measures, the fluorescence intensity (FL) distribution is provided for both positive and negative samples (**Fig. 6c**). The FL values were significantly higher ($P < 0.0001$; two-sided t-test) in patients with IBV than those in controls (**Fig. 6d**). Receiver operating characteristic (ROC) curves were further constructed for patients with IBV. The results show that the diagnostic

accuracy was excellent, with an area under the curve of 1 (**Fig. 6e**). Please note that because of the mandatory mask order in China since 2020 due to COVID-19, we didn't find patients with IAV and all the positive clinical samples (i.e., nasal swab samples) are just for IBV.

The newly added text reads:

POCT application with clinical samples collected on site. To validate FAST-POCT's clinical applicability, we tested 36 clinical samples (nasal swab samples) from patients (n=18) with IBV and control individuals (n=18) without IBV (**Fig. 6a**). Patient information is available in **Supplementary Table 3**. It should be noted that the nasal swab sample collection and the data availability are consented from all participants. IBV infection status was independently confirmed and the study protocols were approved by The First Affiliated Hospital of Zhejiang University (Hangzhou, Zhejiang). Each patient sample was divided into two categories. One aliquot was processed using the FAST-POCT and the other using a benchtop PCR system (SLAN-96P, China). Both assays used the same purification and testing kits. **Figure 6b** shows the results from FAST-POCT and conventional PCR with reverse transcription (RT-PCR). We compared fluorescence intensity (FAST-POCT) with $-\log_2(\text{Ct})$, where Ct is the cycle cut-off of the conventional RT-PCR. A good concordance between these two methods was observed. FAST-POCT and RT-PCR showed a strong positive correlation with the Pearson coefficient (r) values of 0.90 (**Fig. 6b**). We next assessed the diagnostic accuracy of FAST-POCT. As independent analytical measures, the fluorescence intensity (FL) distribution is provided for both positive and negative samples (**Fig. 6c**). The FL values were significantly higher ($P < 0.0001$; two-sided t-test) in patients with IBV than those in control groups (**Fig. 6d**). Receiver operating characteristic (ROC) curves were further constructed for IBV. We found the diagnostic accuracy was excellent, with an area under the curve of 1 (**Fig. 6e**). Please note that because of the mandatory mask order in China since 2020 due to COVID-19, we did not find patients with IAV; as a result, all the positive clinical samples (i.e., nasal swab samples) are only for IBV.

Fig. 6 Testing of clinical samples with FAST-POCT platform for Influenza diagnosis. a, Clinical study design. A total of 36 samples were analyzed using the FAST-POCT platform and the conventional RT-PCR, including 18 patient samples and 18 control individual samples without Influenza. **b**, Evaluation of analytical

concordance between FAST-POCT PCR and conventional RT-PCR. The results were positively correlated (Pearson's $r = 0.90$). **c**, Fluorescence intensity levels for 18 patients with IBV and 18 controls. **d**, FL values were significantly higher in the patients with IBV (+) than in controls (-) (**** $P < 0.0001$; two-sided t-test; $n = 36$). **e**, ROC curves. The dotted line (**d**) denotes the cutoff estimated from the ROC analysis. The AUC was 1 for IBV.

[REDACTED]
Editorial note: image can be
seen in Supplementary Figure 13

Figure S13. Pictures of a, nasal swab collection site, and b, the collected nasal swab samples.

5. The authors should also provide additional results and information to support existing PCR results. In the current manuscript, because the authors used a custom instrument to perform PCR, more details and additional characterization results should be provided. For example, the fluorescence intensities shown in Fig. 5a and Fig. 5b are measured by the detector in the instrument, but what is the detector? Moreover, it would be helpful to visually observe increased fluorescence intensities at the end of PCR. Here, because the PCR chambers have open windows, it is possible to provide fluorescent images before and after PCR. The PCR condition – such as the thermocycling protocol and the primer concentration – should also be provided in much more detail. The reviewer also wondered why there did not seem to be a reverse transcription step in the thermocycling protocol. Based on the results shown in Fig. 5a(ii) and Fig. 5b(ii), the PCR efficiencies can be interpreted as greater than 100%. The authors should share their thoughts about the unexpectedly good efficiencies. Finally, the authors should also briefly mention how they loaded the PCR reagents. Did they inject the PCR reagents before they assemble the cartridge?

We have added more information to address this comment. The customized fluorescence intensity detector is shown in **Fig. S10b**. The fluorescent images with IAV and IBV after 40 cycles are provided in **Figs. 5a** and **5b**, respectively.

Thank you for pointing out the PCR efficiencies greater than 100%. We assume that the pipetting, diluting and PCR reagents issues may result in the abnormal PCR efficiencies. Therefore, we re-calibrated the pipettor, used a new batch of PCR purification and testing kit from the kit supplier (Liferiver, China) and conducted the same experiments as demonstrated in **Fig. 5** again. The new results are shown in the updated **Fig. 5**. It can be seen that the PCR efficiencies are all within the range of 90% ~ 100%, which can meet the PCR testing requirement.

The purification and testing reagent are all pre-loaded after the cartridge assembles through the injection port of the cartridge.

The new text reads:

Figure 5a shows the PCR testing results for IAV using 150 μ l purified viral RNA as samples. **Figure 5a(i)** shows that when the concentration of IAV was 10^6 copies/ml, the fluorescence intensity (ΔR_n) can be 0.830,

and as the concentration decreased to 10^2 copies/ml, ΔRn can still be as high as 0.365, which is about 100-fold higher than that of the blank negative control group (0.002). For quantitative analysis, a linear calibration curve between the log concentration of IAV and the cycle threshold (CT) was obtained (Fig. 5a(ii)) with $R^2 = 0.993$ in the range of 10^2 to 10^6 copies/ml based on six separate experiments. These results agree well with the conventional RT-PCR techniques. Figure 5a(iii) shows the fluorescence images of the testing results after 40 cycles from FAST-POCT platform. We found that the FAST-POCT platform can detect as low as 10^2 copies/ml IAV. However, the conventional method does not have a Ct value in the concentration of 10^2 copies/ml, making its LOD about 10^3 copies/ml. Figure 5b shows the PCR testing results for purified IBV RNA samples, with the concentration ranging from 10^1 to 10^6 copies/ml. The results are similar to the IAV testing, which achieved $R^2=0.994$ and LOD of 10^2 copies/ml.

Fig. 5 PCR testing results for influenza virus. a, PCR testing results for influenza A virus (IAV) with IAV concentration ranging from 10^6 to 10^1 copies/ml using TE buffer as the negative control (NC). (i) real-time fluorescence profile. (ii) linear calibration curve between the log concentration of IAV RNA and cycle threshold (CT) for both FAST and conventional testing technique. (iii) fluorescence images of FAST-POCT with IAV after 40 cycles. b, PCR testing results for influenza B virus (IBV) with (i) real-time fluorescence profile. (ii) the linear calibration curve and (iii) fluorescence images of FAST-POCT with IBV after 40 cycles.

The lower limit of detection (LOD) for IAV and IBV using FAST-POCT platform is 10^2 copies/ml, which is lower than that of the conventional methods (10^3 copies/ml). **c**, Multiple detection results for IAV and IBV. GAPDH was used for positive control, and the TE buffer was used for negative control to prevent possible contamination and background amplification. Four different types of samples are identified: (1) negative sample (“IAV-/IBV-”) with only GAPDH; (2) IAV-infected (“IAV+/IBV-”) with IAV and GAPDH; (3) IBV-infected (“IAV-/IBV+”) with IBV and GAPDH; (4) IAV/IBV-infected (“IAV+/IBV+”) with IAV, IBV and GAPDH. The dash lines in the figures denote the threshold line.

The thermocycling protocol is provided in **Fig. S7**, including the reverse transcription time of 20 min and the thermocycling time of 60 min (one thermocycle time of 90 s).

Figure S7. Temperature profile for the testing process. a, Reverse transcription time is 20 min. Thermocycling time is 60 min. One thermocycling time is 90 s. **b**, Thermal images during PCR testing for one thermocycle, showing fairly stable thermal behavior.

6. The authors should reconsider their claim that FAST-POCT “achieves 10-fold higher sensitive [sic] than the conventional PCR systems”. The authors used different RNA extraction methods for their POC platform and their conventional PCR systems; making a direct comparison between methods with at least two variables (i.e., different extraction and different platform) seems non-ideal. At the very least, the authors must be clear about the differences. The authors may also want to at least compare the extraction method on benchtop and/or just PCR (sans RNA extraction) in FAST-POC versus benchtop instrument so that the improvement from FAST-POC can be better assessed.

Thanks for the great comment regarding the design of the experiment. To address this comment, we used the FAST-POCT platform for RNA extraction and then used the extracted RNA samples to run PCR reactions using FAST-POCT and the benchtop PCR system for comparison. The results are almost identical. A new

figure (Fig. S8) was added.

The new text reads:

It should be noted that we used the FAST-POCT platform for RNA extraction and then used the extracted RNA samples to run PCR reactions using FAST-POCT system and the benchtop PCR system for comparison. The results are almost identical as shown in **Supplementary Figure S8**.

Figure S8. The results for IBV PCR testing using FAST-POCT platform and benchtop platform with RNA samples extracted by the FAST platform.

7. The reviewer would like the authors to improve their reasoning for their observed improvement in FAST-POC. In lines 235 – 236, the authors presume that the improvement “may be due to the small thermal inertia and the high surface-to-volume ratio of the microscale PCR reaction chamber”. The reviewer cannot agree with this reason. Based on the reviewer’s estimate from Fig. S5, the PCR chamber area (the part that is visible to eye) appears comparable to 0.2 mL PCR tubes. The reaction volume in the PCR chamber was 20 μ L, which is on par with typical benchtop PCR. From what the reviewer can see, one possible way that the thermal mass of can be smaller for the PCR in FAST-POC is if the PCR is heated through the thin plastic film rather than the thicker substrate layer – a point that the authors unfortunately did not clarify in the current manuscript. As for the surface area to volume ratio, the reviewer can agree that it is likely slightly higher for the PCR chamber in FAST-POC, but it is also not more than an order of magnitude different, so it is difficult to the reviewer to believe that difference led to more efficient thermocycling.

Thanks for this excellent comment. We have added new figures to more clearly present this point. As shown in **Fig. S10b**, there is a groove on the back of the testing chamber, making the bottom thickness of the testing chamber about 200 μ m and thus leading to a small thermal inertia, which may contribute to the observed improvement in FAST-POCT. It can be seen from **Fig. S7** and **Movie S7** that the cooling process takes about 10 s and the heating process takes about 15 s and the one thermocycle time is within 90 s. The chamber surface is 96.6 mm² and the volume of the chamber is 25 mm³, making the surface-to-volume ratio about 3.86. The PCR testing experiments for purified IAY RNAs were also conducted for evaluating the mixing performance, including shaking mixing (the same mixing method as the conventional RT-PCR operation), bubble mixing (the present method, 3s at the pressure of 0.12 bar) and without mixing as a control group. The results can be found in **Supplementary Figure S12**. It can be seen when the RNA concentration is high (10⁶ copy/ml), the

Ct value for different mixing methods is almost the same with the Ct value of bubble mixing. As the RNA concentration decreases to 10^2 copy/ml, the shaking mixing and the control group exhibit no Ct value while the bubble mixing method still obtains a Ct value of 36.9, which is below the threshold Ct value of 38. The results show the advantage of bubble mixing, which is also demonstrated in the literature (Burger, S. et al. Lab Chip 16, 2016, 219-390) and it may also explain the reason why FAST-POCT platform shows a slightly higher sensitivity than that of the conventional RT-PCR instrument.

The new text related to this point now reads:

The PCR testing process is conducted, and the thermal profile is provided in **Supplementary Figure S7**, including the reverse transcription time of 20 min and the thermocycling time (95°C and 60°C) of 60 min, with one thermocycle for 90 s (**Supplementary Movie S7**). It takes less time for FAST-POCT to complete one thermocycle (90 s) than that of conventional RT-PCR (180 s for one thermocycle). This can be attributed to the high surface-to-volume ratio and the small thermal inertia of the microscale PCR reaction chamber. The chamber surface is 96.6 mm^2 and the volume of the chamber is 25 mm^3 , making the surface-to-volume ratio about 3.86. It can be seen in **Supplementary Figure S10** that there is a groove on the back of the PCR testing area of our platform, making the bottom thickness of the PCR chamber $200\text{ }\mu\text{m}$. A heat-transferring elastic pad is adhered to the heating surface of the temperature control unit, ensuring the tight contact with the back surface of the testing chamber. In this way, the thermal inertia of the platform can be reduced, and the heating/cooling efficiency is increased.

The PCR testing experiments for purified IAY RNAs were conducted to evaluate different mixing methods, including shaking mixing (the same mixing method as the conventional RT-PCR operation), bubble mixing (the present method, 3 s at the pressure of 0.12 bar) and without mixing as a control group. The results can be found in **Supplementary Figure S12**. It can be seen when the RNA concentration is high (10^6 copy/ml), the Ct value for different mixing methods is almost the same with the Ct value of bubble mixing. As the RNA concentration decreases to 10^2 copy/ml, the shaking mixing and the control group exhibit no Ct value while the bubble mixing method still obtains a Ct value of 36.9, which is below the threshold CT value of 38. The results show the advantage of bubble mixing, which is also demonstrated in the literature⁶⁰ and it may also explain the reason why the FAST-POCT platform has a slightly higher sensitivity than that of the conventional RT-PCR.

Figure S12. Comparison of PCR performance using different mixing methods of shaking mixing, bubble mixing (the present work), and without mixing, as a control group.

8. The authors must better substantiate or contextualize the one-hour claim. First, the authors must provide detailed thermocycling protocol and temperature ramping performances of their instrument. Again, based on the layout of the instrument and the reviewer's experience, it seems really difficult to perform rapid thermocycling, even if a smaller heater that only covers the 4 PCR chambers in FAST-POC is used. Also, based on Fig. S8, the authors seem to be able to perform 18 cycles between 95 degrees C and 60 degrees C in ~40 min. The authors seem to perform 10-minute hot start at 95 degrees C, which is a rather lengthy. The authors also seem to have skipped the reverse transcription step. If the authors used a really high concentration of target so that they can perform few PCR cycles so that the total turnaround time is under 60 min, that is OK, but the authors still must clearly state so.

Thanks again for this comment. In the original manuscript, we failed to mention the reverse transcription step though we did have it in the experiment (20 minutes for this step), which is now corrected in the revised manuscript. We also updated the thermocycling protocol in the revised manuscript (**Fig. S7**). Due to the rapid heating/cooling performance (thin bottom thickness of the testing chamber mentioned in comment 3), one thermocycle can be performed within 90 s, making the whole thermocycling time 60 min for 40 cycles. Considering the automatic sample preparation time of about 2 min, the sample-in-answer-out time is about 82 min.

The related new text now reads:

The PCR testing process is conducted, and the thermal profile is provided in **Supplementary Figure S7**, including the reverse transcription time of 20 min and the thermocycling time (95°C and 60°C) of 60 min, with one thermocycle for 90 s (**Supplementary Movie S7**). It takes less time for FAST-POCT to complete one thermocycle (90 s) than that of conventional RT-PCR (180 s for one thermocycle). This can be attributed to the high surface-to-volume ratio and the small thermal inertia of the microscale PCR reaction chamber. The chamber surface is 96.6 mm² and the volume of the chamber is 25 mm³, making the surface-to-volume ratio about 3.86. It can be seen in **Supplementary Figure S10** that there is a groove on the back of the PCR testing area of our platform, making the bottom thickness of the PCR chamber 200 μm. A heat-transferring elastic pad is adhered to the heating surface of the temperature control unit, ensuring the tight contact with the back surface of the testing chamber. In this way, the thermal inertia of the platform can be reduced, and the heating/cooling efficiency is increased.

9. The authors should also improve their discussion section. For example, the authors should discuss areas of improvement for their FAST technology. The reviewer can see that the authors may want to discuss ways to decrease the footprint of their cartridge (and hence their instrument). The authors can also see that the authors may want to discuss how to free their system of pumps and gauges.

Thanks again for another great comment. We have rewritten our discussion section to have an in-depth discussion and perspectives. The new text now reads:

In this paper, we demonstrated a film-lever actuated switch technology (FAST), which possesses the desired features for an ideal POCT. The advantages of our technology include: (1) multifunctional dispensing (in cascaded, simultaneous, sequential, and selective manner), on-demand releasing (rapid and proportional releasing to the applied pressure) and robust operation (without leakage under the vibration of 150 rad/min); (2) long-term storage (accelerated life tests of 2 years with about 0.3% weight loss); (3) capability in handling the liquid with a wide range of wettability and viscosity (viscosity as high as 5,500 cp); (4) cost-effectiveness (estimated material costs for the FAST-POCT PCR device is about \$1). By combining the multifunctional dispensing units, an integrated FAST-POCT platform was demonstrated and applied to PCR testing of

influenza A and B viruses. The IAV and IBV can be detected in situ with 10^2 copies/ml of LOD, and the one-step pathotyping of IAV and IBV on the FAST-POCT platform was achieved in 82 min. The clinical tests with 36 nasal swab samples showed good concordance in fluorescence intensity with standard RT-PCR (Pearson coefficients > 0.9). Parallel to this work, varieties of emerging biochemical techniques (i.e., plasmonic thermocycle testing, amplification-free immunoassay, and nanobody-functionalized testing) have shown their potential for POCT. However, due to a lack of an all-integrated and robust POCT platform, these techniques inevitably require separate pre-treatment processes (e.g., RNA extraction³⁷, incubating³⁸ and rinsing³⁹), which further makes the present work complementary to these technologies to achieve advanced POCT capabilities with desired “sample-in-answer-out” performance. In this work, a pneumatic pump is used to activate the FAST valves, which in principle can be replaced by other means for a smaller form factor, such as use of an electromagnetic force. Further improvements can include, for example, customizing the cartridge for different and specific biochemical assays, and adopting different heating/cooling systems and layouts to reduce the size of the instrument for PCR applications. We believe that the proposed FAST technique represents a novel potential to establish a universal platform not just for biomedical testing, but also for environmental monitoring, food quality inspection, new material synthesis and pharmaceuticals, given that the FAST platform provides a new means to manipulate fluids.

Minor Comments

1. The reviewer thinks that mixing can also be presented as a stand-alone functionality of FAST. In the current manuscript, mixing was only included as part of Fig. 4, which appeared rather suddenly. The authors can consider including a mixing demonstration in Fig. 3 so that future readers can be more oriented for Fig. 4. The authors should consider making this change.

Thank you for your useful suggestion. FAST platform provides fluid manipulation, while mixing is not the key feature of the FAST platform so that we included it in Fig. 4. But we did re-organize the text to make the logic flow better.

The new text reads:

The RNA adsorption steps are shown in **Fig. 4a** and are as follows: (1) The sample was pipetted into the M chamber and released to the mixing chamber when the pressure P_1 ($= 0.26$ bar) was applied. (2) The air pressure P_2 ($= 0.12$ bar) was applied through channel A, which was connected to the bottom of the mixing chamber. Though plenty of mixing techniques show their potentials for liquid mixing in the POCT platforms (e.g., serpentine mixing⁵³, chaotic mixing⁵⁴ and batch-mode mixing⁵⁵), their mixing efficiency and effectiveness are still unsatisfactory. Here, a bubble mixing method is employed, in which air is introduced to the bottom of the mixing chamber generating air bubbles in the liquid; the strong vortex then enables full mixing performance within several seconds. The bubble mixing experiments were carried out and the results are provided in Supplementary Figure S6. It can be seen that when 0.10 bar pressure was applied, it took about 8s to complete a full mixing. As the pressure increased to 0.20 bar, it only took about 2s to obtain a full mixing. The method to calculate the mixing efficiency is provided in the method section.

2. The authors should include one or two sentences in the caption of Fig. 2 to describe terminologies in Fig. 2c (i.e., B, q, Fs, Tx). The reviewer saw that these terms were defined in the text, but it would make digesting the information easier if these terms were also defined and contextualized in the caption.

Thank you for your useful suggestion. We added the description of the terminologies (i.e., B, q, Fs, Tx). in Fig. 2C. Now the caption reads:

Fig. 2 Mechanism and theoretical analysis of FAST device. Schematic diagrams and pictures for **a**, Sealed state. When no pressure is applied, the levers press the film to the blocks and the liquid is sealed. **b**, Opened state. When pressure is applied, the film is expanded and pushes the lever up, thus, the channel opens, and the liquid can flow. **c**, Characteristic sizes that determine the critical pressure. Characteristic sizes involve the length of the lever (L), the distance between the block and the hinge (l) and the protrusion thickness of the lever (t). F_s is the sealing force at the block point B. q is a uniformly distributed load on the lever. T_x denotes the torque at the hinge generated by the lever. Critical pressure means the pressure needed to push the lever up and make the liquid flow. **d**, The theoretical and experimental results for the relationship between the critical pressure and the characteristic sizes.

0. The authors should improve the clarity of “T-shaped clip structure”, which were mentioned in lines 92 and 96. The authors have many options here. They can add one or two descriptive sentences, label either Fig. 1 or Fig. 2, refer to Fig. S1, or do a combination of these. This said, the reviewer thinks it is unnecessary to describe the clip structure for assembling the cartridge. So, the authors can also remove the information from the main text and only include it in the supplementary information.

Thanks for the suggestion. We added some description sentences in the “Design, mechanism and theoretical analysis of FAST-POCT platform” part for T-shape clip structure in the supplementary information.

The new text reads:

The T-shape clip has a clearance between two legs. When the clip is pushed into the groove, the two legs bend slightly and then recover to its original state as it comes through the groove and tightly binds the cover and the substrate (**Supplementary Figure S1**).

1. The authors should provide more experimental details to the “Setup for FAST performance tests” section in Methods. For example, how was the liquid volume in each chamber calculated and then presented in Fig. 3? Did the authors capture a video using their high-speed camera (should provide vendor information), take still images at fixed time points (using what software), calculate the area and then multiply by the depth of the chamber?

More details were added. A fixed amount of liquid was injected into the testing device and a high-speed camera was applied to record the flow behavior. After that, the still images were captured from the flow behavior video

at fixed time points and the remaining area was calculated using software Image-Pro Plus and then multiplied by the depth of the chamber.

Related text now reads:

Setups for FAST performance tests. The pictures of the multifunctional dispensing, on-demand release and robust operations were all taken using a high-speed camera (Sony AX700 with 1000fps). The orbital shaker used in the robustness tests was purchased from SCILOGEX (SCI-O180). An air compressor was used to generate the air pressure, and several digital precision pressure regulators were used to adjust the value of the pressure. The flow behavior testing process is the following. A given amount of liquid was injected into the testing device and a high-speed camera was applied to record the flow behavior. The still images were then captured from the flow behavior video at fixed time points and the remaining area was calculated using software Image-Pro Plus and then multiplied by the depth of the chamber to calculate the volume. Details on the flow behavior testing system can be found in **Supplementary Figure S4**.

For the second reviewer:

The paper presents a new switching mechanism for liquid handling, in particular for point of care applications. The authors demonstrate a sample-in answer-out PCR based analysis using their switch technology. The paper is well written and the structure is clear.

1. The switch provides an interesting technological approach for liquid handling. From my perspective, the impact and the scientific value of the presented work, however, falls short for a publication in Nature Communication. Point-of-Care nucleic acid analysis has been discussed in numerous papers and there are products on the market with the most famous one being probably the Cepheid GeneXpert system.

As the authors mention, centrifugal microfluidics is one of the most promising candidates. Here, also products for sample in answer out testing are on the market:

Meridianbioscience with the revogene:

<https://www.meridianbioscience.com/diagnostics/platforms/molecular/revogene/?country=US>.

Spindiag with the Rhonda system:

<https://www.spindiag.de/spindiag-rhonda-pcr-rapid-covid-19-test-watch/>

We acknowledge the development of the POCT technologies, and all the great contributions made by the previous researchers and commercial efforts to put this technology to the market. To the best of our knowledge, however there are only limited microfluidic technologies on the market that *simultaneously* meet the desired performance, including multifunctional dispensing, on-demand release, robust operations, long-term storage, high-viscosity liquid handling, and cost-effective fabrication. The uniquely comprehensive capability of our device, we believe has exciting potential to break down current barriers for mass adoption of POCT products (e.g., Cepheid, Binx, Visby, Cobas and Liat) in the open market. To the best of our ability and based on available data we have tried to analyze the sensitivity of existing commercial platforms. While statistical information is not readily available beyond the dimensions/specifications of these products, below is a quick synopsis of what we learned.

Cepheid GeneXpert system may be the most successful POCT platform, accounting for about 25% of the nucleic acid analysis using the POCT platform in US. A syringe motor is used to drive the liquid into the cartridge and a valve driving motor is used to rotate the bottom of the cartridge to change the injection of different reagents. However, the cartridge comprises too many components, such as cartridge body, syringe barrel, rotary valve, pistol, gear, filter, and so on, and it takes more than five steps to assemble the device, which increases the complexity and fabrication cost (about \$17~35 per test). Our platform is comprised of only four components (estimated material cost of \$1), and it takes two steps to accomplish the assembly process.

The Revogene and Rhonda are two successful POCT products using centrifugal microfluidic technology (i.e., CD-based products). The cartridges of these products contain all reagents necessary to directly detect the pathogens (i.e., SARS-CoV-2, or other viral and bacterial pathogens) and several analyses for the same pathogens can be automatically analyzed in one run within one hour. Because of the intrinsic problems of centrifugal microfluidic technology (i.e., only outward fluid transportation), these products cleverly bypass the comprehensive fluid manipulation by adopting purification-free technology, which targets particular infectious disease diagnosis and unfortunately may not a general solution to developing a platform for a wider range of applications and use-case scenarios (Clime, L. et al. *Microfluidics and Nanofluidics*, 2019, 23:29). A comprehensive fluid manipulation would be thus desired for general purpose.

Compared with these technologies, the present FAST platform has the following advantages:

- a) Multifunctional dispensing (in cascaded, simultaneous, sequential, and selective manner), on-demand releasing (rapid and proportional to the applied pressure) and robust operation (without leakage under the vibration of 150 rad/min).
- b) Long-term storage (accelerated life tests of 2 years with about 0.3% weight loss).
- c) Capability in handling the liquid with a wide range of wettability and viscosity (viscosity as high as 5,500 cP).
- d) Cost-effective (estimated material costs for the FAST-POCT PCR cartridge is about \$1, Supplementary Table 1).

Therefore, we believe that our FAST platform does open a new opportunity for POCT system.

2. *The authors claim that forces directing radially outwards is a significant challenge in centrifugal microfluidics but this challenge has been solved. Clime et al. Lab Chip, 2015, 15, 2400-2411 discusses an active approach for liquid control. Zehnle et al. presented a fully passive approach (Lab Chip, 2012, 12, 51425145). The passive approaches with no extra means have been applied by Czilwik et al. (Lab Chip, 2015, 15, 3749-3759) or Rombach et al. (Analyst, 2020, 145, 7040-7047) for sample-in answer out of nucleic acid analysis with pre-stored reagents.*

Thanks for pointing out these references. The Zengerle group makes a great contribution to the CD-based POCT system. Several references are included in our original manuscript, such as the miniature stick-packaging method (Lab Chip, 2013, 13, 2888-2892) and the batch-mode mixing method (Lab Chip, 2005, 5, 560-565). We did miss these important references regarding using pneumatic operation on the CD to drive the liquid inwards and we added and analyzed these references in our revised manuscript.

The new text reads:

Combining pneumatic operations in centrifugal microfluidics, e.g., the centrifugo-dynamic method³⁷⁻³⁹ and the active pneumatic method⁴⁰ have shown to be an appealing alternative. In the centrifugo-dynamic method, an additional cavity and connecting microchannels are integrated in the device to enable both outward and inward manipulations, though its pumping efficiency (ranging from 75% to 90%) depends highly on number of pumping cycles as well as the viscosity of the fluids. In the active pneumatic method, on-demand fluid releasing and inward manipulation are achieved by simultaneously applying positive pressure and precisely matched rotational speed, enabled by a high-speed motor. There have been other successful methods that employ only pneumatic driving mechanisms (positive^{41,42} or negative pressure⁴³) and a normally closed valve structure. By applying pressure sequentially in the pneumatic chamber, the liquid is pumped forward in a peristaltic way by which the normally closed valve avoids the liquid from flowing back, thereby enabling complex fluid manipulations. To the best of our knowledge, however, currently there are only limited microfluidic technologies that can perform complex fluid handling in a single POCT device, which includes multifunctional dispensing, on-demand release, robust operation, long-term storage, high-viscosity liquid handling, and cost-effective fabrication – all at the same time. A lack of multi-step functionality operating also could be one of the reasons why only a few commercial POCT products (e.g., Cepheid, Binx, Visby, Cobas Liat and Rhonda) to date have been successfully adopted in the open market.

A more detailed comparison of the performance between the existing POCT technologies and our method is provided in Supplementary Table 2. To make the comparison clearer for the papers mentioned by this reviewer, the related papers are underlined in this response letter, though they are not underlined in the revised manuscript.

Supplementary Table 2. Comparisons among some POCT PCR studies and technologies.

Detection performance						
Reference	FAST-POCT (this work)	Gzilwik, et al, Lab Chip, 15, 3749-3759 (2015)	Clime, et al, Lab Chip, 15, 2400-2411 (2015)	Wang, et al, Biosens. Bioelectron, 41, 484-491 (2013)	Rombach, et al, Analyst, 145, 7040-7047 (2020)	Jung, et al, Biosens. Bioelectron, 68, 218-224 (2015)
Target	Influenza RNA	Bacterial pathogens DNA	DNA	HIV DNA	Respiratory tract infection pathogens	Influenza A RNA
Method	RT-PCR	RT-PCR	Only DNA extraction	RT-PCR	RT-PCR	Isothermal PCR
Detection limit	15 copies	200 cfu of S. agalactiae	⁻¹	62 copies	-	10 copies
Testing time	82 min	225 min	-	95 min	200 min	45 min
Fluid handling capability						
Multifunctional dispensing	Cascaded, simultaneous, sequential and selective dispensing	Inward dispensing	Inward dispensing	-	Inward dispensing	Two-way dispensing
On-demand releasing	Rapid and proportional to the applied pressure	-	Yes	Yes	-	-
Robust operation	No leaking under the vibration of 150 rad/min	-	-	-	-	-
Long-term storage	Accelerated life tests of DI water with weight loss less than 0.3% for two years	-	-	-	Yes	-
Liquid properties	Manipulating the liquid with viscosity as high as 5,500 cP	Viscosity of 16 cP with efficiency of 75%	-	-	-	-
Driving mechanism	Positive pressure	Centrifugal force	Centrifugal force and positive pressure	Negative pressure	Centrifugal force	Centrifugal force

¹ The dash “-” means that the related results or features were not demonstrated in the paper.

3. Taking these publications into considerations, even though new, I do not see sufficient impact for the presented technology and not enough novelty for the discussed application that justifies publication in *Nature Communications*. I do think the technology is valuable and in summary worth to be published. I thus recommend that the authors select a journal that is more specific to the topic.

As we have addressed in comments 1 and 2 for this reviewer, the present FAST platform does provide sufficient new features to resolve the limitations by other existing methods in a substantial way. Elaborations and practical demonstrations (e.g., using clinical samples) were included in the revised manuscript. We hope that our revised manuscript is persuasive.

4. The authors claim that the presented RT-PCR is more sensitive than a conventional RT-PCR system. I could not find a reasonable explanation and in particular not a discussion on why this increase of sensitivity could be related to their technology, thus, such a statement is misleading from my perspective. The claim: “This may be due to the small thermal inertia and the high surface-to-volume ratio of the microscale PCR reaction chamber, as well as the high efficiency of the bubble mixing” is highly speculative. For example, if you compare bubble mixing with the used mixing by hand, this may be the case, then, why did the authors not use Vortex mixing? I would assume that if one optimizes the assay procedure for the reference system, a similar sensitivity would be reached.

Thank you for your comment. We have added new figures to more clearly present this point. As shown in Fig. S10b, there is a groove on the back of the testing chamber, making the bottom thickness of the testing chamber about 200 μm and thus leading to a small thermal inertia, which may contribute to the observed improvement in FAST-POCT. It can be seen from **Fig. S7** and **Movie S7** that the cooling process takes about 10 s and the heat process takes about 15 s and the one thermocycle time is within 90 s. The chamber surface is 96.6 mm^2 and the volume of the chamber is 25 mm^3 , making the surface-to-volume ratio about 3.86. The PCR testing experiments for purified IAV RNAs were also conducted to evaluate the mixing performance, including shaking mixing (the same mixing method as the conventional RT-PCR operation), bubble mixing (the present method, 3s at the pressure of 0.12 bar), and without mixing as a control group. The results can be found in **Supplementary Figure S12**. It can be seen when the RNA concentration is high (10^6 copy/ml), the Ct value for different mixing methods is almost the same with the Ct value of bubble mixing. As the RNA concentration decreases to 10^2 copy/ml, the shaking mixing and the control group exhibit no Ct value while the bubble mixing method still obtains a Ct value of 36.9, which is below the threshold Ct value of 38. The results show the advantage of bubble mixing, which is also demonstrated in the literature (Burger, S. et al. *Lab Chip* 16, 2016, 219-390) and it may also explain the reason why FAST-POCT platform shows a slightly higher sensitivity than that of the conventional RT-PCR instrument.

The new text related to this point now reads:

The PCR testing process is conducted, and the thermal profile is provided in **Supplementary Figure S7**, including the reverse transcription time of 20 min and the thermocycling time (95°C and 60°C) of 60 min, with one thermocycle for 90 s (**Supplementary Movie S7**). It takes less time for FAST-POCT to complete one thermocycle (90 s) than that of conventional RT-PCR (180 s for one thermocycle). This can be attributed to the high surface-to-volume ratio and the small thermal inertia of the microscale PCR reaction chamber. The chamber surface is 96.6 mm^2 and the volume of the chamber is 25 mm^3 , making the surface-to-volume ratio about 3.86. It can be seen in **Supplementary Figure S10** that there is a groove on the back of the PCR testing area of our platform, making the bottom thickness of the PCR chamber 200 μm . A heat-transferring elastic pad is adhered to the heating surface of the temperature control unit, ensuring the tight contact with the back surface

of the testing chamber. In this way, the thermal inertia of the platform can be reduced, and the heating/cooling efficiency is increased.

The PCR testing experiments for purified IAY RNAs were conducted to evaluate different mixing methods, including shaking mixing (the same mixing method as the conventional RT-PCR operation), bubble mixing (the present method, 3s at the pressure of 0.12 bar) and without mixing as a control group. The results can be found in **Supplementary Figure S12**. It can be seen when the RNA concentration is high (10^6 copy/ml), the Ct value for different mixing methods is almost the same with the Ct value of bubble mixing. As the RNA concentration decreases to 10^2 copy/ml, the shaking mixing and the control group exhibit no Ct value while the bubble mixing method still obtains a Ct value of 36.9, which is below the threshold CT value of 38. The results show the advantage of bubble mixing, which is also demonstrated in the literature⁶⁰, and it may also explain the reason why the FAST-POCT platform has a slightly higher sensitivity than that of the conventional RT-PCR.

Figure S12. Comparison of PCR performance using different mixing method, including shaking mixing, bubble mixing (the present work) and without mixing as a control group.

For the third reviewer:

The manuscript by Liang and colleagues describes a mechanical valving technique that is used to control liquids flow in microfluidic devices. The mechanism provides an incremental improvement compared with similar techniques found in the literature. The introduction is rather superficial, and the claims are exaggerated.

Thanks for the comment, which motivates us to present a much clearer presentation. We have rewritten the introduction section to analyze advantages and disadvantages of the existing POC technologies (e.g., centrifuge-dynamic pumping method, pneumatic-assisted centrifugal method and pneumatic with normally closed valve method). We also made a comparison between the performance of the existing ones and our technology (**Supplementary Table 2**). To the best of our knowledge, in spite of excellent contributions made by the previous researchers, to date, there are very few microfluidic technologies that simultaneously meet the desired performance, including multifunctional dispensing, on-demand release, robust operations, long-term storage, high-viscosity liquid handling, and cost-effective fabrication, all of which currently present barriers to mass adoption of POCT products (e.g., Cepheid, Binx, Visby, Cobas Liat and Rhonda) in the open market.

The next text for the introduction section reads:

Combining pneumatic operations in centrifugal microfluidics, e.g., the centrifuge-dynamic method³⁷⁻³⁹ and the active pneumatic method⁴⁰ have shown to be an appealing alternative. In the centrifuge-dynamic method, an additional cavity and connecting microchannels are integrated in the device to enable both outward and inward manipulations, though its pumping efficiency (ranging from 75% to 90%) depends highly on number of pumping cycles as well as the viscosity of the fluids. In the active pneumatic method, on-demand fluid releasing and inward manipulation are achieved by simultaneously applying positive pressure and precisely matched rotational speed, enabled by a high-speed motor. There have been other successful methods that employ only pneumatic driving mechanisms (positive^{41,42} or negative pressure⁴³) and a normally closed valve structure. By applying pressure sequentially in the pneumatic chamber, the liquid is pumped forward in a peristaltic way by which the normally closed valve avoids the liquid from flowing back, thereby enabling complex fluid manipulations. To the best of our knowledge, however, currently there are only limited microfluidic technologies that can perform complex fluid handling in a single POCT device, which includes multifunctional dispensing, on-demand release, robust operation, long-term storage, high-viscosity liquid handling, and cost-effective fabrication – all at the same time. A lack of multi-step functionality operating also could be one of the reasons why only a few commercial POCT products (e.g., Cepheid, Binx, Visby, Cobas Liat and Rhonda) to date have been successfully adopted in the open market.

The Supplementary Table 2 compares the performance of the existing POC technologies and the present one. **Supplementary Table 2. Comparisons among some POCT PCR studies and technologies.**

Detection performance						
Reference	FAST-POCT (this work)	Gzilwik, et al, Lab Chip, 15, 3749-3759 (2015)	Clime, et al, Lab Chip, 15, 2400-2411 (2015)	Wang, et al, Biosens. Bioelectron, 41, 484-491 (2013)	Rombach, et al, Analyst, 145, 7040-7047 (2020)	Jung, et al, Biosens. Bioelectron, 68, 218-224 (2015)
Target	Influenza RNA	Bacterial	DNA	HIV DNA	Respiratory	Influenza A

		pathogens DNA			tract infection pathogens	RNA
Method	RT-PCR	RT-PCR	Only DNA extraction	RT-PCR	RT-PCR	Isothermal PCR
Detection limit	15 copies	200 cfu of S. agalactiae	†	62 copies	–	10 copies
Testing time	82 min	225 min	–	95 min	200 min	45 min
Fluid handling capability						
Multifunctional dispensing	Cascaded, simultaneous, sequential and selective dispensing	Inward dispensing	Inward dispensing	–	Inward dispensing	Two-way dispensing
On-demand releasing	Rapid and proportional to the applied pressure	–	Yes	Yes	–	–
Robust operation	No leaking under the vibration of 150 rad/min	–	–	–	–	–
Long-term storage	Accelerated life tests of DI water with weight loss less than 0.3% for two years	–	–	–	Yes	–
Liquid properties	Manipulating the liquid with viscosity as high as 5,500 cP	Viscosity of 16 cP with efficiency of 75%	–	–	–	–
Driving mechanism	Positive pressure	Centrifugal force	Centrifugal force and positive pressure	Negative pressure	Centrifugal force	Centrifugal force

† The dash “-” means that the related results or features were not demonstrated in the paper.

2. While the functionality of the method is explained well in some cases the authors do not explain the source of the pneumatic pressure. A successful device should not require expensive, bulky instrument, on contrary it has to be operated without any external source to apply pressure or actuate the system. The functionality of the device should be investigated with liquids with different viscosities and mechanical properties and under various climate conditions. The details of how the external pressure is applied are not presented. The dependency on the use of external pressure for opening and closing does not make the POCT an ideal solution for the POCT test.

Thanks for the comment. We agree with the reviewer that a successful POCT system should not require

expensive, bulky instrument. Figure S10 shows that one testing module is rather small with $L \times W \times H = 138 \times 109 \times 134$ mm. The final instrument can have one or several such modules. The price of the main component is provided in Supplementary Table 1, which indicates that the cartridge cost is fairly low.

Figure S10. Illustration of the FAST-POCT testing module. **a**, The demonstration of the way to place the FAST-POCT device into a testing module prototype. **b**, i) The schematic diagram of the PCR testing unit. ii) Picture of the front view of the testing module prototype. iii) Picture of the back view of the testing module prototype. **c**, Flow chart of the relationship of different components. The scale bars are 10 mm.

Supplementary Table 1. Material cost estimate of the FAST-POCT system

Instrument (main components)			
Description	Supplier	Size	Cost
Air pump	Zhirong, China Air flow: 10L/min Positive pressure: >1.5bar	L×W×H: 106×51×84 mm	\$13.12
Electromagnetic valve	Highend, China Working pressure: 0~7bar Response time: 10ms	L×W×H: 82.5×39×25 mm	\$33.41
Electromagnet	Elecall, China Maximum force: 15kg	Φ30×22 mm	\$3.13
Temperature control unit	Yexian, China Accuracy: ±0.1°C	L×W×H: 40×40×25 mm	\$123.25

PCR testing unit	Customized Excitation: 470nm Emission: 510nm	L×W×H: 50×37×32 mm	\$75.47
Fast-POCT device (for one person)			
Description	Supplier	Size	Cost
Substrate ¹	Ender, China	L×W×H: 120×95×5 mm	\$0.28
Cover ¹	Ender, China	L×W×H: 120×95×2 mm	\$0.19
Elastic film ²	Dow Corning, Slygard 184, USA	L×W×H: 120×95×0.35 mm	\$0.38
Plastic film	Saiweige, China	L×W×H: 120×95×0.1 mm	\$0.01
Adhesive film	Adhesive research, 90880, USA	L×W×H: 120×95×0.14 mm	\$0.47
Purification kit ³	Liferiver, Z-ME-0010, China	N/A	\$1.25
Testing kit ³	Liferiver, RR-0051-02/RR- 00202China	N/A	\$2.00
Estimated total material cost of a FAST-POCT kit for multiplexed influenza tests			\$5.20

¹ The substrate and cover are fabricated using 3D printing techniques in the PCR testing experiments and the cost can be reduced in the future when the mass-production techniques are applied like mold-injection.

² The elastic film used in the experiments are made from slygard 184 using mold-casting techniques. Further reduction of the cost should be possible in the future if commercially available elastic films are applied.

³ The purification and testing kits are purchased from supplier for experimental purposes. When the FAST-POCT device is in the market and the kit can be purchased in a large scale, the kit cost can be further reduced.

We conducted additional experiments to address the viscosity issue. The new text now reads:

The capability of handling liquids with different wettability and viscosity in a same device is still challenging for the POCT application. Low wettability may cause leaking or other unintentional flowing behavior in the channel and the preparation of high viscosity liquid often requires auxiliary instrument, such as vortex mixers, centrifuges, and strainers⁵². We tested the relationship between the critical pressure and the liquid properties (with a wide range of wettability and viscosity). The results are shown in **Table 1** and **Movie S5**. It can be seen that the liquid with different wettability and viscosity can all be sealed tightly in the chamber, and when the pressure is applied, even the liquid with the viscosity as high as 5,500 cP can also be transferred the next chamber, which makes the high viscosity sample testing (i.e., sputum, a highly viscous specimen used to diagnose respiratory disease) possible.

Table 1 The relationship between liquid properties and critical pressure

Description	Acetone	Ethanol	Methanol	DI water	Glycerin	Sylgard 184
Contact angle (°)	17.2	22.7	25.3	87.2	45.6	62.3
Viscosity (cP)	0.3	1.2	0.8	1.0	1500.0	5500.0
Critical pressure (bar)	0.116	0.128	0.121	0.154	0.218	0.317

3. *The results are compared with existing liquid handling techniques, this makes it impossible for the reader to see the advantage/disadvantage of their technique contrast especially when compared with the liquid handling methods that are already commercialized e.g., cobas® Liat®. The authors claim that film-lever actuated switch technology provides full control in liquid handling. However, regarding the materials used, fabrication technique, and complexity of their design, the repeatability, and the robustness of their method necessitates more experiments with different liquids and in different environments. The POCT must be able to perform in extreme climate conditions and therefore the 0.8% liquids loss 60 days at room temperature is remarkably high. This is a common drawback of storing liquids directly in a diagnostic platform, which is why aluminum, glass, and micro-dispenser containers are applied.*

Appreciate this excellent comment. Cobas Liat features a tube-like cartridge with several packages containing all the necessary reagents. The Cobas Liat PCR System is small, easy to use and fast—able to provide testing results by the end of a patient’s appointment. However, the cartridge is not capable of multifunctional dispensing and all the packages of the cartridge are connected in a cascaded manner where the reagents are released sequentially by applying pressure. This kind of liquid manipulation may result in the reagent contamination and cause false positive according to a warning from FDA (Potential for False Results with Roche Molecular Systems, Inc. cobas SARS-CoV-2 & Influenza Test for use on cobas Liat System-Letter to Clinical Laboratory Staff, Point-of-Care Facility Staff, and Health Care Providers, Letters to Health Care Providers).

To the best of our knowledge, till now there are still limited microfluidic technologies which can achieve multifunctional dispensing, on-demand release, robust operations, long-term storage, high-viscosity liquid handling and cost-effective fabrication at the same time. This is maybe the reason why although varieties of POCT products emerge (e.g., Cepheid, Binx, Visby, Cobas Liat and Rhonda), very few of them are widely applied in the open market.

The advantages of our technology can be summarized as follows:

- (1) Multifunctional dispensing (in cascaded, simultaneous, sequential, and selective manner), on-demand releasing (rapid and proportional to the applied pressure) and robust operation (without leakage under the vibration of 150 rad/min).
- (2) Long-term storage (accelerated life tests of 2 years with about 0.3% weight loss).
- (3) Capability in handling the liquid with varieties of wettability and viscosity (viscosity as high as 5500 cP).
- (4) Cost-effective (estimated material costs for the FAST-POCT PCR device is about \$1, Supplementary Table 1).

Really appreciate this reviewer’s point about liquid storage performance. We re-examined the present liquid storage performance and found a way to significantly improve it. In the original manuscript, we demonstrated this performance using 3D printed materials with a pretty rough surface (Fig. S5) with roughness about 25 microns. Such roughness may not affect the liquid flow behavior but would affect the long-term storage performance since the liquid may be evaporated through the small clearance. In the revised manuscript, we have changed the 3D printed material to a PMMA material fabricated by the CNC technique. We tested the long-term storage capability by conducting accelerated life tests. The testing devices were filled with DI-water and 70% ethanol at 65 °C for 9 days. We used Arrhenius equation and the activation energy for permeation reported in the literature to calculate the equivalent real time. Figure 3b shows that the average weight loss of five samples maintained at 9 days at 65 °C, i.e., equivalent to 0.30% for DI water and 0.72% for 70% ethanol for over 2 years at 23 °C. We now can consider this as an excellent liquid storage performance.

Figure S5. Images of the surface roughness comparison of a, 3D printing material and b, PMMA material.

Figure 3b. The results for the long-term storage tests of DI water and 70% Ethanol. The new text now reads:

Long-term reagent storage is another essential characteristic for a successful POCT device, which will allow untrained personnel to handle multiple reagents. Although many techniques show their potential for long-term storage (e.g., micro-dispenser³⁵, blister⁴⁸ and stick packages⁴⁹), a special receiving chamber is required to hold the packages, thereby increasing cost and complexity; moreover, these storage mechanisms do not enable on-demand releasing and lead to loss of reagents due to residuals in the packages. The capability of long-term storage was tested by conducting accelerated life tests, using PMMA material fabricated by the CNC technique due to its small roughness and the resistance to gas permeation (**Supplementary Figure S5**). The testing devices were filled with DI-water (deionized water) and 70% ethanol (to simulate the volatile reagents) at 65 °C for 9 days. Both DI water and ethanol were stored using an aluminum foil to seal their top entrance. An Arrhenius equation and the activation energy for permeation reported in the literature^{50, 51} were applied for

calculating the equivalent real time. **Figure 3b** shows the results the average weight loss of five samples maintained at 9 days at 65 °C, i.e., equivalent to 0.30% for DI water and 0.72% for 70% ethanol for over 2 years at 23 °C.

4. The technology developed in this study is potentially interesting for POCT, however, there are several limitations. The authors claimed the medical relevance of their device, but they miss the point of diagnostic POCT. There is a clear misalignment between the technology development and the actual diagnostic capability of their device. To be successful a diagnostic technology should complete the cycle from "sampling to treatment", but the paper lacks the whole treatment, usability, and user-interaction of their potential POCT.

Thanks for the comment and we have conducted real patient test using the present FAST-POCT unit. To validate FAST-POCT's clinical applicability, we tested 36 clinical samples (nasal swabs samples) from patients (n=18) with IVB and control individuals (n=18) without IVB (**Fig. 6a**). Patient information is provided in **Supplementary Table 3**. IBV infection status was independently confirmed by The First Affiliated Hospital of Zhejiang University (Hangzhou, Zhejiang). Dr. Jun Fan (a clinical physician) from this hospital was responsible for sample collection and independent verification. Each patient sample was divided into two. One aliquot was processed using the FAST-POCT and the other using a benchtop PCR system (SLAN-96P, China). Both assays used the same purification and testing kits. **Figure 6b** shows the results from FAST-POCT and conventional PCR with reverse transcription (RT-PCR). We compared raw fluorescence intensity (FAST-POCT) with $-\log_2(\text{Ct})$, where Ct is the cycle cut-off of the conventional RT-PCR. A good concordance between these two methods was observed. FAST-POCT and the conventional RT-PCR showed a strong positive correlation with the Pearson coefficient (r) values of 0.90 (**Fig. 6b**). We next assessed the diagnostic accuracy of FAST-POCT. As independent analytical measures, the fluorescence intensity (FL) distribution is provided for both positive and negative samples (**Fig. 6c**). The FL values were significantly higher ($P < 0.0001$; two-sided t-test) in patients with IBV than those in controls (**Fig. 6d**). Receiver operating characteristic (ROC) curves were further constructed for patients with IBV. The results show that the diagnostic accuracy was excellent, with an area under the curve of 1 (**Fig. 6e**). Please note that because of the mandatory mask order in China since 2020 due to COVID-19, we didn't find patients with IAV and all the positive clinical samples (i.e., nasal swab samples) are just for IBV.

The newly added text reads:

POCT application with clinical samples collected on site. To validate FAST-POCT's clinical applicability, we tested 36 clinical samples (nasal swab samples) from patients (n=18) with IVB and control individuals (n=18) without IVB (**Fig. 6a**). Patient information is available in **Supplementary Table 3**. It should be noted that the nasal swab sample collection and the data availability are consented from all participants. IBV infection status was independently confirmed and the study protocols were approved by The First Affiliated Hospital of Zhejiang University (Hangzhou, Zhejiang). Each patient sample was divided into two categories. One aliquot was processed using the FAST-POCT and the other using a benchtop PCR system (SLAN-96P, China). Both assays used the same purification and testing kits. **Figure 6b** shows the results from FAST-POCT and conventional PCR with reverse transcription (RT-PCR). We compared fluorescence intensity (FAST-POCT) with $-\log_2(\text{Ct})$, where Ct is the cycle cut-off of the conventional RT-PCR. A good concordance between these two methods was observed. FAST-POCT and RT-PCR showed a strong positive correlation with the Pearson coefficient (r) values of 0.90 (**Fig. 6b**). We next assessed the diagnostic accuracy of FAST-POCT. As independent analytical measures, the fluorescence intensity (FL) distribution is provided for both positive and negative samples (**Fig. 6c**). The FL values were significantly higher ($P < 0.0001$; two-sided t-test) in patients with IBV than those in control groups (**Fig. 6d**). Receiver operating characteristic (ROC) curves were further constructed for IBV. We found the diagnostic accuracy was excellent, with an area under the curve of 1 (**Fig.**

6e). Please note that because of the mandatory mask order in China since 2020 due to COVID-19, we did not find patients with IAV; as a result, all the positive clinical samples (i.e., nasal swab samples) are only for IBV.

Fig. 6 Testing of clinical samples with FAST-POCT platform for Influenza diagnosis. **a**, Clinical study design. A total of 36 samples were analyzed by FAST-POCT platform and the conventional RT-PCR, including 18 patients' samples and 18 control individuals without Influenza. **b**, Evaluation of analytical concordance between FAST-POCT PCR and conventional RT-PCR. The results were positively correlated (Pearson's $r = 0.90$). **c**, Fluorescence intensity levels for 18 patients with IBV and 18 controls. **d**, FL values were significantly higher in the patients with IBV (+) than in controls (-) (**** $P < 0.0001$; two-sided t-test; $n = 36$). **e**, ROC curves. The dotted line (**d**) denotes the cutoff estimated from the ROC analysis. The AUC was 1 for IBV.

[REDACTED]

Editorial note: image can be seen in Supplementary Figure 13

Figure S11. Pictures of a, nasal swab collection site and b, the collected nasal swab.

Regarding the “sampling to treatment” as this reviewer mentioned, we have to admit that we now focus on “sampling to diagnostics”, and treatment is totally a different story.

5. *Justifying the diagnostic-amenability by a proof-of-concept PCR test for influenza virus is not enough for justifying the POCT potential. I suggest the author resize their medical relevant claims.*

We have conducted clinical sample tests as discussed in the previous comment to justify the POCT potential, tuned down our medical relevant claims, such as removing “10-folds higher sensitivity”, and added more discussions to provide perspective. The new text now reads:

In this paper, we demonstrated a film-lever actuated switch technology (FAST), which possesses the desired features for an ideal POCT. The advantages of our technology include: (1) multifunctional dispensing (in cascaded, simultaneous, sequential, and selective manner), on-demand releasing (rapid and proportional releasing to the applied pressure) and robust operation (without leakage under the vibration of 150 rad/min); (2) long-term storage (accelerated life tests of 2 years with about 0.3% weight loss); (3) capability in handling the liquid with a wide range of wettability and viscosity (viscosity as high as 5,500 cp); (4) cost-effectiveness (estimated material costs for the FAST-POCT PCR device is about \$1). By combining the multifunctional dispensing units, an integrated FAST-POCT platform was demonstrated and applied to PCR testing of influenza A and B viruses. The IAV and IBV can be detected in situ with 10^2 copies/ml of LOD, and the one-step pathotyping of IAV and IBV on the FAST-POCT platform was achieved in 82 min. The clinical tests with 36 nasal swab samples showed good concordance in fluorescence intensity with standard RT-PCR (Pearson coefficients > 0.9). Parallel to this work, varieties of emerging biochemical techniques (i.e., plasmonic thermocycle testing, amplification-free immunoassay, and nanobody-functionalized testing) have shown their potential for POCT. However, due to a lack of an all-integrated and robust POCT platform, these techniques inevitably require separate pre-treatment processes (e.g., RNA extraction³⁷, incubating³⁸ and rinsing³⁹), which further makes the present work complementary to these technologies to achieve advanced POCT capabilities with desired “sample-in-answer-out” performance. In this work, a pneumatic pump is used to activate the FAST valves, which in principle can be replaced by other means for a smaller form factor, such as use of an electromagnetic force. Further improvements can include, for example, customizing the cartridge for different and specific biochemical assays, and adopting different heating/cooling systems and layouts to reduce the size of the instrument for PCR applications. We believe that the proposed FAST technique represents a novel potential to establish a universal platform not just for biomedical testing, but also for environmental monitoring, food quality inspection, new material synthesis and pharmaceuticals, given that the FAST platform provides a new means to manipulate fluids.

6. *They fail to cite the right diagnostic literature. Citations 1,2,3,4 should be complemented with the right relevant report from e.g., the WHO (when cited) or analyses of diagnostic potentials from a health system relevance (e.g., Madhukar Pai research work). They re-cite statements from other papers, without actually providing the right original reference.*

Thanks for this great comment. We have revised the manuscript to reflect this point. New references have been cited.

1. WHO Coronavirus (COVID-19) Dashboard. Available at: www.covid19.who.int.
2. Chagla, Z. & Madhukar, P. COVID-19 boosters in rich nations will delay vaccines for all. *Nat. Med.* **27**, 1659-1665 (2021).
3. Faust, L. et al. SARS-CoV-2 testing in low- and middle-income countries: availability and affordability in the private health sector. *Microbes Infect* **22**, 511-514 (2020).
4. World Health Organization. Global prevalence and incidence of selected curable sexually transmitted infections: overview and estimates. Geneva: WHO, WHO/HIV_AIDS 2, (2001).

7. After implementing correction, the article would be more suitable for a technical journal, as it is a proof-of-concept type of study. Such liquid handling technology has potential relevance, however claiming its diagnostic potential is too preliminary at the stage the device is.

The present FAST platform does provide sufficient new features to resolve the limitations by other existing methods in a substantial way. Elaborations and practical demonstrations (e.g., using clinical nasal swab samples, and a device unit presentation) were included in the revised manuscript. We assume that our work is more than a proof-of-concept type of study. In fact, it clearly demonstrates the diagnostic potential of our platform. We hope that our revised manuscript is persuasive.

REVIEWER COMMENTS

Reviewer #1 (Remarks to the Author):

The reviewer applauds the authors for their considerable efforts in preparing the extensively revised (and improved) manuscript, which fully addresses the reviewer's comments.

Reviewer #2 (Remarks to the Author):

The revised manuscript has addressed all my technical concerns in a sufficient manner. The authors have made the potential much more clear in comparison to the previous version. There is no doubt that the approach is worth to be published in general. Nevertheless, I see the impact of the presented technology as medium and from my perspective, it is more appropriate for a journal focusing on technical improvements.

Reviewer #3 (Remarks to the Author):

Dear authors,

Thanks for adding important information. I regretfully believe that the presented manuscript is not suitable for Nature communications; however, it will be appreciated in journals with a narrower scope as it provides significant improvements in the related field.

The presented literature review seems to highlight handpicked techniques and lacks a comprehensive comparison. For example, an article by M. Aeinshvand (Lab Chip, 2015, 15, 3358-3369; Biosensors and Bioelectronics, Volume 67, 2015, Pages 424-430) uses a membrane-based pneumatic and a thermo-pneumatic that uses a similar concept. This will make it hard for the readers to clearly see the benefit of FAST method, in comparison with the others.

Specifically, I cannot agree with the author's opinion regarding portable storage devices (blisters and micro-dispenser). The need for an external chamber is not a strong argument, especially for the micro-dispenser by Kazemzadeh et al., as it can be manufactured in a much smaller dimension than the presented mechanism by the authors. The cost of portable devices can be lower as they can be easily adapted to various platforms. The portable devices are easier to apply and can be applied on linear and non-linear platforms, showing their wider applications. Table 11, listing the instrument, shows that the device requires several additional equipment compared to a micro-dispenser or blisters; as such, the overall price of the device can be higher.

For the first reviewer:

The reviewer applauds the authors for their considerable efforts in preparing the extensively revised (and improved) manuscript, which fully addresses the reviewer's comments.

Appreciate your supportive comments.

For the second reviewer:

The revised manuscript has addressed all my technical concerns in a sufficient manner. The authors have made the potential much more clear in comparison to the previous version. There is no doubt that the approach is worth to be published in general. Nevertheless, I see the impact of the presented technology as medium and from my perspective, it is more appropriate for a journal focusing on technical improvements.

Thank you for your supportive comments.

For the third reviewer:

Thanks for adding important information. I regretfully believe that the presented manuscript is not suitable for Nature communications; however, it will be appreciated in journals with a narrower scope as it provides significant improvements in the related field.

The presented literature review seems to highlight handpicked techniques and lacks a comprehensive comparison. For example, an article by M. Aeinehvand (Lab Chip, 2015,15, 3358-3369; Biosensors and Bioelectronics, Volume 67, 2015, Pages 424-430) uses a membrane-based pneumatic and a thermo-pneumatic that uses a similar concept. This will make it hard for the readers to clearly see the benefit of FAST method, in comparison with the others.

Thanks for the comment. We carefully studied the references mentioned in the comment. In the thermo-pneumatic method (Lab Chip, 2015,15, 3358-3369), a latex membrane and a liquid transition chamber were specially designed to seal or reopen an inlet when a trapped air volume is heated or cooled. However, the heating/cooling setup brings about a slow actuation issue and restrict its usage in thermal-sensitive assay (e.g., PCR amplification). In the other reference (Biosensors and Bioelectronics, Volume 67, 2015, Pages 424-430), a special microballoon structure was integrated in the CD-like device to facilitate the mixing performance with no demonstration for other features. The FAST-POCT platform is capable of completing the fully mixing performance in 2 s with a pressure of 0.20 bar (Figure S6) and simultaneously with the features of (1) multifunctional dispensing (in cascaded, simultaneous, sequential, and selective manner), on-demand releasing (rapid and proportional releasing to the applied pressure) and robust operation (without leakage under the vibration of 150 rad/min); (2) long-term storage (accelerated life tests of 2 years with about 0.3% weight loss); (3) capability in handling the liquid with a wide range of wettability and viscosity (viscosity as high as 5,500 cp); (4) cost-effectiveness (estimated material costs for the FAST-POCT PCR device is about \$1).

The new text for the introduction section reads:

In the thermo-pneumatic method, a latex membrane and a liquid transition chamber were specially designed to seal or reopen an inlet when a trapped air volume is heated or cooled. However, the heating/cooling setup brings about a slow actuation issue and restrict its usage in thermal-sensitive assay (e.g., PCR amplification).

Specifically, I cannot agree with the author's opinion regarding portable storage devices (blisters and micro-dispenser). The need for an external chamber is not a strong argument, especially for the micro-dispenser by Kazemzadeh et al., as it can be manufactured in a much smaller dimension than the presented mechanism by the authors. The cost of portable devices can be lower as they can be easily adapted to various platforms. The portable devices are easier to apply and can be applied on linear and non-linear platforms, showing their wider applications. Table 11, listing the instrument, shows that the device requires several additional equipment compared to a micro-dispenser or blisters; as such, the overall price of the device can be higher.

The micro-dispenser in a smaller dimension by Kzaemzadeh et al. can only deal with the simple one-step assay process, such as the finger-actuated manipulation and the blood plasma separation. However, when dealing with complex assay with multiple steps, the micro-dispenser is integrated to a CD-like device. As we have stated in the “Long-term reagent storage” part, a special receiving chamber for the CD-like device is required to hold the packages, thereby increasing cost and complexity. Additionally, the liquid below the aperture of the micro-dispenser may not be fully transported, leading to residuals in the micro-dispenser. The CD-like device integrated with the micro-dispenser also require additional high-speed motor to drive the liquid and temperature control system to perform the PCR detection (however, the author did not demonstrate such applications for their micro-dispenser platform). In our work, all the chambers are integrated in the FAST-POCT platform with no need of separate holding or connecting parts for each chamber and no liquid residual issues, which can be readily mass-fabricated using mold-injection technique.